# DNA sequence and chromatin modifiers cooperate to confer epigenetic bistability at imprinting control regions

Stefan Butz[1,2], Nina Schmolka[1,9], Ino D. Karemaker [1], Rodrigo Villaseñor [1,10], Isabel Schwarz [1], Silvia Domcke[3,4,11], Esther C. H. Uijttewaal[5], Julian Jude[6], Florian Lienert[3,4], Arnaud R. Krebs [3,12], Nathalie P. de Wagenaar[7], Xue Bao[7], Johannes Zuber [6,8], Ulrich Elling [5], Dirk Schübeler [3,4] & Tuncay Baubec [1,7] ✉

Genomic imprinting is regulated by parental-specific DNA methylation of imprinting control regions (ICRs). Despite an identical DNA sequence, ICRs can exist in two distinct epigenetic states that are memorized throughout unlimited cell divisions and reset during germline formation. Here, we systematically study the genetic and epigenetic determinants of this epigenetic bistability. By iterative integration of ICRs and related DNA sequences to an ectopic location in the mouse genome, we first identify the DNA sequence features required for maintenance of epigenetic states in embryonic stem cells. The autonomous regulatory properties of ICRs further enabled us to create DNA-methylation-sensitive reporters and to screen for key components involved in regulating their epigenetic memory. Besides DNMT1, UHRF1 and ZFP57, we identify factors that prevent switching from methylated to unmethylated states and show that two of these candidates, ATF7IP and ZMYM2, are important for the stability of DNA and H3K9 methylation at ICRs in embryonic stem cells.

Epigenetic regulation of gene activity depends on multiple layers of chromatin modifications that are maintained during DNA replication[1,2]. By definition, these epigenetic mechanisms act independently of the DNA sequence at the genomic sites they occupy. However, several studies have highlighted a contribution of DNA sequence to the regulation and maintenance of chromatin modifications, preventing a clear distinction between epigenetic and genetic control of gene activity[3–8]. Genomic imprinting is an epigenetic phenomenon, where DNA methylation marks on either the maternal or paternal ICRs dictate parental-specific activity of transcripts in *cis*[9–11]. ICRs inherit parental-specific DNA methylation marks from either the oocyte or sperm, which are then propagated in all somatic tissues of the next generation[9]. The inheritance of differential epigenetic states on the parental chromosomes, despite identical DNA sequence, identical

[1]Department of Molecular Mechanisms of Disease, University of Zurich, Zurich, Switzerland. [2]Molecular Life Science PhD Program of the Life Science Zurich Graduate School, University of Zurich and ETH Zurich, Zurich, Switzerland. [3]Friedrich Miescher Institute for Biomedical Research, Basel, Switzerland. [4]Faculty of Science, University of Basel, Basel, Switzerland. [5]Institute of Molecular Biotechnology Austria (IMBA), Vienna BioCenter (VBC), Vienna, Austria. [6]Research Institute of Molecular Pathology (IMP), Vienna BioCenter (VBC), Vienna, Austria. [7]Division of Genome Biology and Epigenetics, Institute of Biodynamics and Biocomplexity, Department of Biology, Science Faculty, Utrecht University, Utrecht, the Netherlands. [8]Medical University of Vienna, Vienna BioCenter (VBC), Vienna, Austria. [9]Present address: Institute of Experimental Immunology, University of Zurich, Zurich, Switzerland. [10]Present address: Division of Molecular Biology, Biomedical Center Munich, Ludwig-Maximilians-University, Munich, Germany. [11]Present address: Department of Genome Sciences, University of Washington, Seattle, WA, USA. [12]Present address: European Molecular Biology Laboratory (EMBL), Genome Biology Unit, Heidelberg, Germany. ✉e-mail: t.baubec@uu.nl

chromosomal location and exposure to the same regulatory factors in the nucleus, make ICRs a great model to study the individual contribution of DNA sequence and chromatin modifications to epigenetic memory.

Several factors and mechanisms have been identified that regulate the maintenance of DNA methylation at ICRs. Once methylation marks have been deposited in the germline[12], the maintenance methyltransferase DNMT1 and its accessory protein UHRF1 are responsible for the maintenance of methylation during DNA replication[13]. In addition, several factors have been identified to regulate H3K9me3 at the DNA-methylated ICRs, including SETDB1, KAP1 and G9A[14–16]. Importantly. the KRAB zinc-finger factor ZFP57 binds the methylated hexanucleotide DNA sequence TGCmCGC and recruits KAP1 and other associated factors to establish a feedback between DNA methylation and H3K9me3 at ICRs[16,17]. Indeed, binding of ZFP57 and recruitment of KAP1 are crucial steps in regulating imprints, as knockout (KO) of *Zfp57* in mice results in loss of almost all imprints and embryonic lethality[18–20], and ZFP57 is required for maintenance of DNA methylation and H3K9me3 at ICRs in cellular systems[16,18,21].

Although the factors that control DNA and histone methylation at ICRs have been widely investigated, the DNA sequence properties of ICRs have not been explored in detail. Furthermore, it is also not known if additional key players contribute to the epigenetic maintenance at ICRs. By iterative integration of ICR DNA sequences to the same genomic site in mouse embryonic stem cells (mESCs), we show that ICRs are autonomous genetic elements that can recapitulate the epigenetic states observed at the endogenous locations. Using this setup, we show that by presetting DNA methylation, we can establish two opposing epigenetic states that are faithfully propagated by the ectopic ICR. This DNA-methylation-dependent switch is unique to ICRs. Systematic integrations of variant and synthetic ICRs allowed us to identify the sequence requirements that are necessary and sufficient for this switch-like behavior. Furthermore, by using the ectopic ICRs as DNA-methylation-sensitive reporters in loss-of-function genetic screens, we confirm DNMT1, UHRF1 and ZFP57 as the core epigenetic regulators of genomic imprinting. In addition, we identify ATF7IP and ZMYM2 as factors involved in regulating maintenance of epigenetic states at ICRs.

## Results

### Autonomous ICRs memorize preestablished epigenetic states

We hypothesized that the DNA sequence of ICRs should contain sufficient information to establish and maintain the distinct epigenetic states observed on the parental alleles (Extended Data Fig. 1a). We selected four ICRs from the *Airn*, *Kcnq1ot1*, *Zrsr1* and *H19* imprinting clusters and used recombinase-mediated cassette exchange (RMCE[3]) to integrate them individually into the genome of mESCs (Fig. 1a). To mimic the differential DNA methylation states of the ICRs, we performed RMCE in parallel for unmethylated ICRs and ICRs that were premethylated by the bacterial CpG methyltransferase M.SssI (Fig. 1a and Extended Data Fig. 1b). As a control sequence, we used the *Igf2r* DMR (differentially methylated region), a promoter that acquires differential DNA methylation only during differentiation[22]. Furthermore, we included a set of inactive gene promoters (*Hes3*, *Tcl1* and *Syt1*), which were previously shown to be protected from de novo DNA methylation when integrated to the same RMCE site[3] (Fig. 1b).

After successful integration, we measured DNA methylation at the RMCE site by bisulfite conversion PCR (bsPCR). All four ICRs maintained their preestablished DNA methylation status at the ectopic site, although in some cases, minor de novo methylation at the unmethylated ICRs was observed (Fig. 1b,c and Extended Data Fig. 1c–e). In contrast, maintenance of preestablished DNA methylation was not observed for the *Igf2r* DMR and the control promoter elements (Fig. 1b and Extended Data Fig. 1f–i). The differential DNA methylation states at the ectopic *Airn* ICR were stably maintained after prolonged cultivation

of mESCs for more than 20 passages, or upon integration to a different RMCE position in the genome, and also following in vitro differentiation of mESCs to neuronal progenitors (Extended Data Fig. 2a–c). Furthermore, DNA methylation of the ectopic *Airn* ICR was still retained at high levels after cultivation of mESCs in 2i medium for 10 days, despite the global reduction in 5-methylcytosine resulting from acquiring a naïve stem cell state[23–26] (Extended Data Fig. 2a,d).

Besides DNA methylation, endogenous ICRs further display differential histone modifications (Extended Data Fig. 1a), whereby the methylated ICR is decorated by H3K9me3 and the unmethylated ICR by H3K4me2 (refs. [14,22,27]). We performed chromatin immunoprecipitation (ChIP) quantitative PCR (qPCR) for H3K9me3 and H3K4me2 and compared the enrichment of these marks at the RMCE integrations of the *Airn* and *Kcnq1ot1* ICRs with their endogenous counterparts (Fig. 1d). The unmethylated ICRs at the RMCE site showed lack of H3K9me3 and increased H3K4me2, whereas the premethylated ICRs revealed the opposite pattern, with increased H3K9me3 and absence of H3K4me2 (Fig. 1d). Previous studies identified the DNA-methylation-specific KRAB-Znf protein ZFP57 to be required for maintenance of DNA methylation and H3K9me3 at endogenous ICRs[16,18,21]. This regulation is recapitulated at the ectopic ICR, as CRISPR-Cas9 deletion of *Zfp57* in mESCs results in rapid and complete loss of DNA methylation at both ectopic and endogenous sites (Extended Data Fig. 2e).

### Epigenetic bistability depends on DNA sequence

We set out to test if the ICR DNA sequence is required for epigenetic memory. First, we aimed to identify if smaller ICR fragments would also efficiently memorize preset DNA methylation patterns and repeated the same experiments with four smaller fragments from the *Airn* ICR (Fig. 2a and Extended Data Fig. 2f). None of the tested fragments could faithfully recapitulate the differential methylation maintenance. The same was observed for the paternally methylated *H19* ICR (Extended Data Fig. 2g,h). Previous studies focusing on non-ICR regulatory regions (promoters, CpG islands or enhancers) have revealed that CpG density, GC content and/or nucleotide sequence can influence establishment of DNA methylation patterns[3–6]. Based on their CpG density and GC content, the ICRs tested here are in the range of genomic elements overlapping with unmethylated CpG island promoters (Extended Data Fig. 3a). To investigate if the CpG density and GC content of the ICRs contribute to the maintenance of methylated and unmethylated states, we selected four genomic regions that are highly similar to the *Airn* ICR in size, GC%, CpG number and distribution (Extended Data Fig. 3b–d). These 'Airn'-like elements failed to maintain the differential methylation and adopted a hypomethylated state like their endogenous counterpart, suggesting that DNA sequence length, CpG density and GC content are not sufficient to establish two distinct epigenetic states (Extended Data Fig. 3e,f).

To further distinguish the direct requirement of DNA sequence from CpG and GC content, we generated a synthetic DNA element based on the *Airn* ICR sequence, where we permutated the inter-CpG DNA sequences until 78% mismatch was reached (Extended Data Fig. 4a,b). Importantly, the permutation of the original sequence retained the local GC content and the number and position of the original CpGs. This replacement removed all inter-CpG DNA sequence information, allowing us to distinguish the contribution of DNA sequence from CpG frequency and distribution. We repeated the RMCE experiments with this 'shuffled' *Airn* ICR and observed that it failed to maintain the preset epigenetic state (Fig. 2b). In both cases, DNA methylation reached an intermediate value of 40.3% for the unmethylated and 23.5% for the premethylated insertion, with disordered methylation patterns (Fig. 2b). The sequence alterations further led to reduced establishment of the H3K9me3 and H3K4me2 at the RMCE site, independently of the preset methylation state (Fig. 2c).

The shuffling of the inter-CpG DNA sequence in the *Airn* ICR disrupted all ZFP57 binding motifs, which might explain the observed

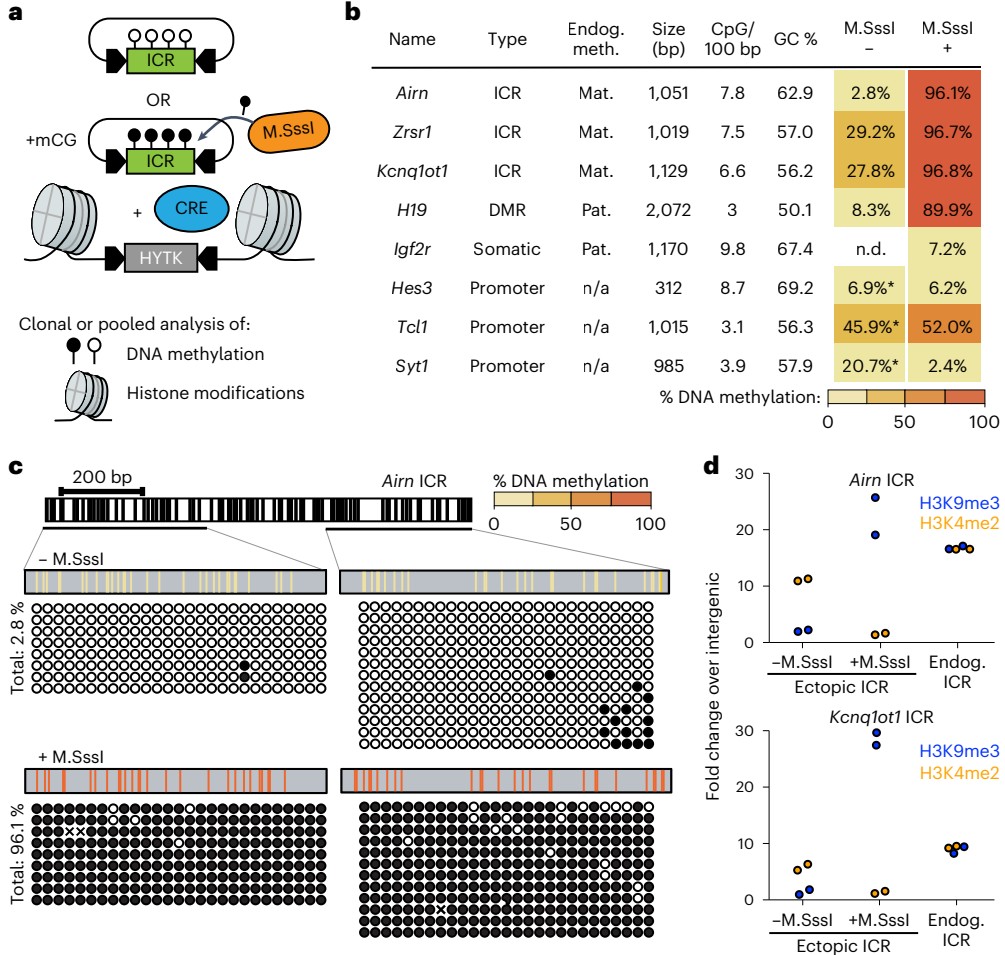

**Fig. 1 | Ectopic ICR sequences recapitulate chromatin states of endogenous ICRs. a,** Experimental overview of stable cell line generation with methylated or unmethylated donor plasmids using RMCE. **b,** Tabular summary of methylation analysis for all integrated ICRs, control DMR and promoter sequences. Endogenous methylation (Endog. meth.) describes the methylation state of the endogenous locus in mESCs. Mat., maternal methylation; Pat., paternal methylation; n/a, no methylation. Size, CpG density (in 100 bp) and GC content (%) are indicated. Total methylation percentages of DNA sequences integrated via RMCE measured by bsPCR is shown for experiments using unmethylated (−M.SssI) and premethylated (+M.SssI) donor plasmids. n/d, not determined. Asterisks indicate measurements obtained from Lienert et al.[3]. **c,** Detailed

methylation analysis for the ectopic *Airn* ICR. CpG positions within the *Airn* ICR sequence are indicated with black vertical lines. Amplified regions for bsPCR are depicted, and single-molecule measurements are shown as black circles corresponding to methylated CpG dinucleotides and white circles to unmethylated CpG dinucleotides. CpG positions marked with 'x' correspond to unaligned nucleotides due to sequencing errors. Aggregated methylation values are displayed as color-coded vertical lines at the respective CpG position. **d,** ChIP-qPCR measurements at ectopic and endogenous ICRs compared to an intergenic site. H3K9me3, blue; H3K4me2, orange. Data points indicate individual technical replicates.

lack of maintenance, in agreement with an in silico evaluation of ZFP57 binding to wild-type and shuffled *Airn* ICR sequence using BPNet[28] (Extended Data Fig. 4c,d). Accordingly, we wanted to investigate if ZFP57 motifs are sufficient for the maintenance of the epigenetic state. Therefore, we restored the ZFP57 binding motifs in the shuffled ICR (Extended Data Fig. 4e) and introduced this methylated and unmethylated DNA element to the RMCE site in mESCs. Although the unmethylated version failed to maintain the hypomethylated state, the premethylated ICR was able to maintain a fully hypermethylated state (Fig. 2d). Given these observations, we wondered if the requirement for ZFP57 binding sites is dependent on the cellular context, especially as *Zfp57* gene expression is tissue specific[18,29]. Therefore we introduced the shuffled *Airn* ICR to RMCE-competent mouse erythroleukemia (MEL) cells[30] and performed targeted bisulfite sequencing. Both methylated and unmethylated shuffled ICRs retained the preset DNA methylation patterns (Fig. 2e), indicating that in MEL cells, CpG content is sufficient for the memory of DNA methylation states.

## Ectopic ICRs establish epigenetic silencing in *cis*

Endogenous ICRs are *cis*-regulating elements that dictate the allelic expression of nearby transcripts based on their DNA methylation state[9]. We first wanted to test if ICR sequences can silence three different reporter constructs in presence of DNA methylation when integrated together to the RMCE site (Fig. 3a and Extended Data Fig. 5a). We selected three commonly used constitutive promoters (pCAGGS, hEF1alpha and hPGK) and showed that they can maintain expression of a GFP reporter at the RMCE integration site in absence of ICRs (Extended Data Fig. 5b). Next, we measured the ability of three methylated ICRs (*Airn*, *Kcnq1ot1* and *Peg10*) to stably repress these promoters at the same RMCE integration site (Fig. 3b). All tested ICR sequences showed stable repression in combination with the Ef1alpha and hPGK promoters. In contrast, the methylated promoters without ICRs, or in combination with the *Dazl* promoter, which is known to be regulated in a DNA-methylation-dependent manner[31], were not able to maintain a repressed state (Fig. 3b). The synthetic pCAGGS promoter gave varying results, depending on the used ICR, suggesting that the strength of this

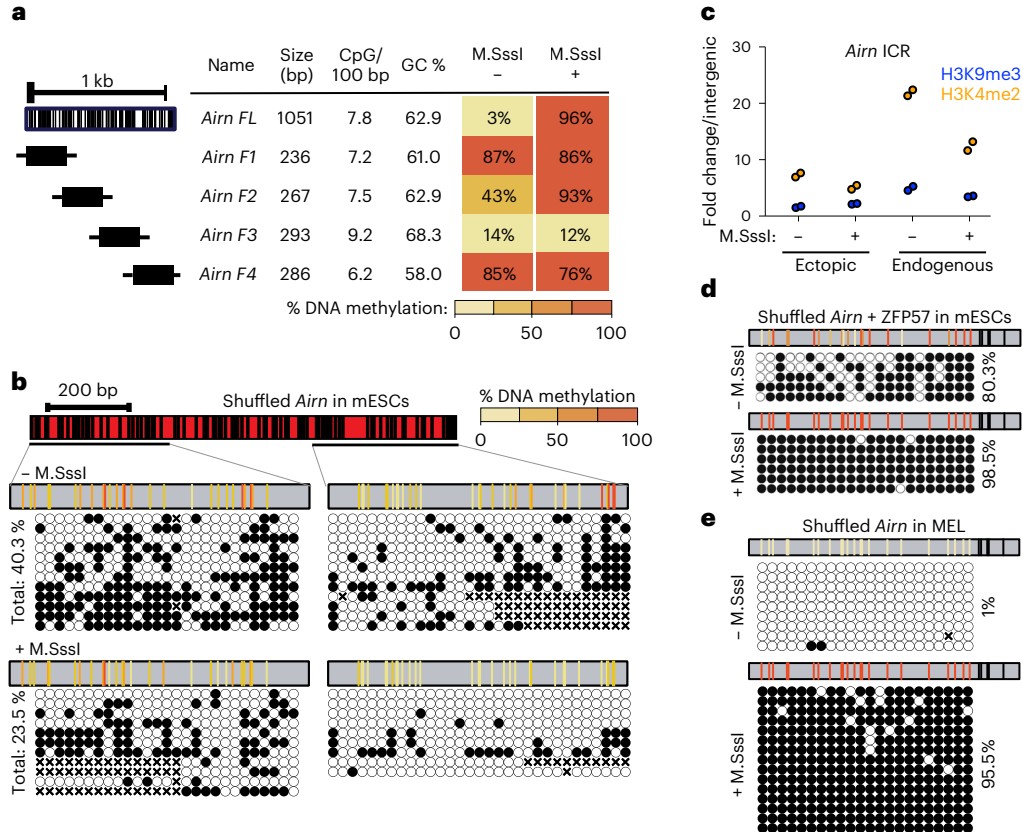

**Fig. 2 | Maintenance of epigenetic memory at ICRs is sequence dependent in mESCs. a**, Tabular summary of methylation analysis for all *Airn*-ICR fragments schematically indicated on the left. In addition, fragment length, CpG densities and GC content is shown for each fragment. Same representation as in Fig. 1b. **b**, Methylation analysis for the shuffled *Airn* ICR. Same representation as in Fig. 1c. CpG positions marked with 'x' correspond to unaligned nucleotides due to sequencing errors. **c**, ChIP-qPCR measurements at the shuffled *Airn* ICR at the

ectopic site compared to the endogenous *Airn* ICR. Data points show individual technical replicates. H3K9me3, blue; H3K4me2, orange. **d**, Methylation analysis for the shuffled *Airn* ICR with reconstituted ZFP57 binding sites. Same representation as in panel **b**. **e**, Methylation analysis for the shuffled *Airn* ICR integrated to murine erythroleukemia (MEL) cells shows maintenance of DNA methylation.

composite promoter can overcome the epigenetic repression induced by some ICRs (Fig. 3b). The DNA-methylation-dependent repression was maintained over longer periods, as measured by GFP activity in multiple clonally derived populations after 16, 23 and 30 days (Extended Data Fig. 5c). The same methylated ICR-dependent repression was observed for the paternally methylated *H19* ICR (Extended Data Fig. 5d).

This setup allowed us to test the contribution of DNA methylation and sequence on the silencing potential of ICRs. For this, we made use of the *Airn*-pEF1a-GFP reporter construct that showed stable maintenance of GFP expression when inserted unmethylated and stable silencing when inserted methylated (Fig. 3c). When we replaced the *Airn* ICR with the shuffled *Airn* version, we observed loss of silencing in most of the measured clones already after 16 days and even more after prolonged cultivation, suggesting that methylation-dependent silencing in *cis* requires an intact ICR sequence (Fig. 3c). Finally, we introduced the shuffled *Airn* sequence containing reconstituted ZFP57 binding sites. Although the unmethylated version led to stochastic loss of transcriptional activity, the methylated construct gave rise to stable repression of the nearby promoter for multiple generations, indicating that ZFP57 binding is not only required for the maintenance of epigenetic memory at ICRs but also sufficient for epigenetic silencing in *cis* (Fig. 3c).

To test if the DNA methylation of ICRs is required for the repression of the nearby promoter, we challenged the established reporter cell lines by culturing them in 2i and 2i + vitamin C media. Both conditions reduce genome-wide DNA methylation levels[23–25], whereas addition of vitamin C results in further removal of DNA methylation from ICRs and

repetitive elements[26,32]. GFP repression was maintained in 2i medium; however, repression was progressively lost in presence of 2i + vitamin C (Extended Data Fig. 5e–g). To further test the dependency on DNA methylation for maintaining the repressed state at the ICR reporters, we performed KO experiments of the general DNA methylation maintenance factors *Uhrf1* and *Dnmt1* (ref. [33]). As expected, removal of DNA methylation in these KO cells led to a reactivation of the ICR reporter within 7 days (Extended Data Fig. 6a,b). The low percentage of cells that show GFP reactivation in these assays is due to low KO efficiency in the CRISPR-targeted pool of cells. Therefore, we cultured the *Airn*-ICR reporter in presence of the DNMT1 inhibitor GSK-3484862 (ref. [34]) for 2 days. We observed complete reactivation with over 95% of cells expressing GFP (Fig. 3d and Extended Data Fig. 6c). The use of this DNMT1 inhibitor further allowed us to test if the reactivation is reversible; therefore, we removed GSK-3484862 from the medium and continued cultivation for 7 more days after washout (Fig. 3d and Extended Data Fig. 6c). We observed no resilencing of activated reporters, indicating that once the ICR is switched on, it cannot revert to a silent state.

## CRISPR screens identify regulators of epigenetic memory at ICRs in mESCs

After establishing multiple ICR-specific reporter cell lines, we wanted to screen for proteins required for maintenance of repressive ICR states. We first established the CRISPR screen workflow using a targeted library against 1,051 chromatin-related factors with 6,204 guide

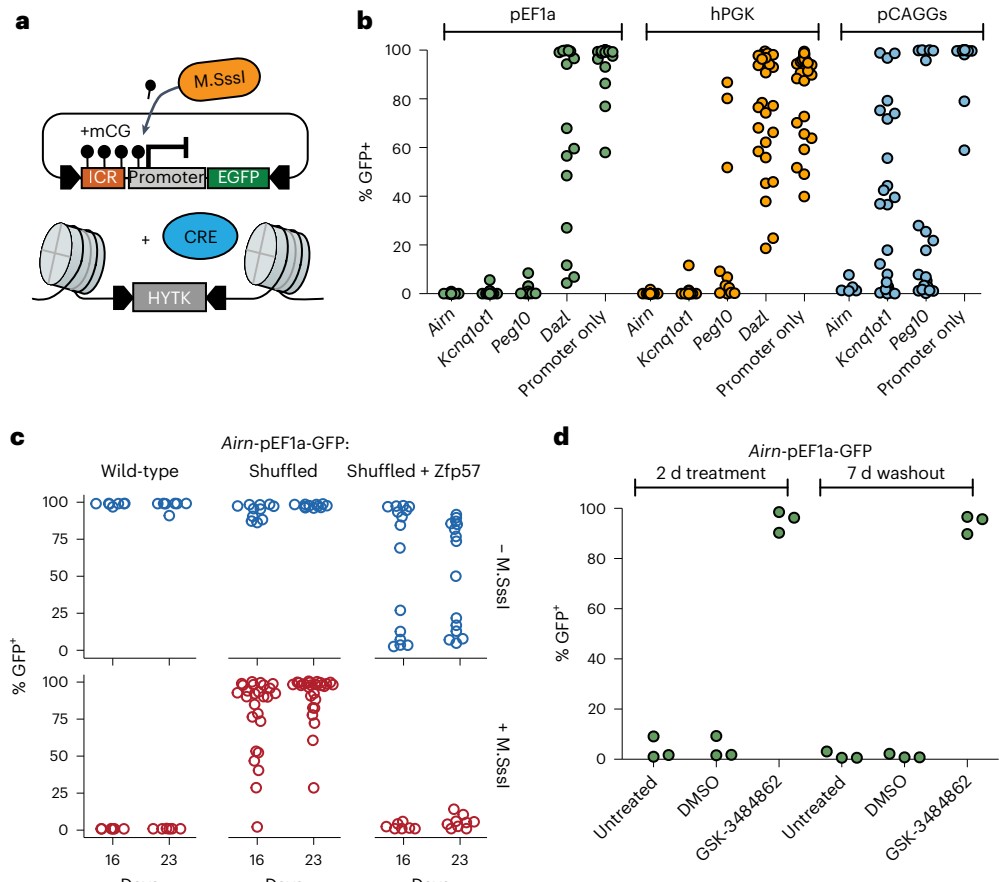

**Fig. 3 | Methylated ICR sequences repress nearby promoters in a sequence-dependent manner. a**, Schematic and experimental overview of reporter cell line generation using RMCE. **b**, Flow cytometric analysis of GFP expression 12 days after transfection with different premethylated ICR/promoter combinations. Each data point shows percentage of GFP-positive cells measurement from a clonally derived cell population. **c**, Flow cytometric analysis indicates percentage of GFP-positive cells in independent cell lines retrieving methylated or unmethylated RMCE donor plasmids containing the wild-type, shuffled *Airn* ICR or the shuffled *Airn* ICR with reconstituted ZFP57 sites in combination with the pEF1a promoter. GFP activity was measured at two consecutive time points (16 and 23 days). **d**, Flow cytometric analysis of three independent clones with the methylated *Airn*-CAG reporter after 2 days treatment with the DNA methylation inhibitor GSK-3484862 and untreated and DMSO controls. Measurements were repeated 7 days after washout of the drug to test for reversion of the reporter silencing. Same representation as in panel b.

RNAs (ChromMM library) and a control library with 500 non-targeting guides in the pCAGGS-*Airn* reporter cell line (Fig. 4a and Extended Data Fig. 7a), and we determined the time point to collect positive clones (Extended Data Fig. 7b). We performed the screen in three methylated ICR reporter lines (*Airn*, *Kcnq1ot1* and *Peg10*) and collected GFP-positive cells after 8 days and repeated the screen for *Airn*, *Kcnq1ot1* in sensitized 2i medium conditions (Extended Data Fig. 7c,d).

As expected, the three positive controls *Zfp57*, *Uhrf1* and *Dnmt1* scored as the top hits in all screens (Fig. 4b,c, Extended Data Fig. 8a–c and Supplementary Table 1). Additionally, other heterochromatin-associated factors like *Cbx1*, *Cbx5*, *Atrx*, *Daxx* and *Setdb1* were enriched in the GFP-positive fraction. The list of high-confidence hits that were repeatedly found in all screens was enriched for *Zfp57*, *Uhrf1* and *Dnmt1*, whereas other hits were identified in individual ICR reporter cell lines (Fig. 4c). We redesigned an extended CRISPR library (EpiTF) consisting of 20,470 guide RNAs against 4,095 genes encoding nuclear factors to cover a large fraction of the KRAB zinc-finger protein family and repeated the screen using the *Airn* ICR reporter (Supplementary Table 1). Despite the increased complexity of the library, we did not identify additional transcription factors to play a role in the maintenance of *Airn* reporter silencing (Fig. 4c and Extended Data Fig. 8d,e). Several candidates identified in more than

one screen were tested by single-KO validation. *Zfp57*, *Uhrf1* and *Dnmt1* showed consistent upregulation in all three reporter lines, whereas other candidates resulted in lower or stochastic reactivation in some of the tested reporter lines (Extended Data Fig. 8f).

## ATF7IP and ZMYM2 colocalize to endogenous ICRs

Two factors were identified in at least three different screens (Fig. 4c and Extended Data Fig. 8g): ATF7IP, responsible for SETDB1-mediated silencing of transposable elements[35–37], as well as ZMYM2, an ATF7IP-interacting factor associated with growth restriction of human pluripotent cells[38,39]. Given their association with H3K9me3 and reported involvement in transcriptional silencing of repetitive elements, we tested their contribution to regulation of epigenetic maintenance at ICRs. In addition human ATF7IP was recently identified to be a repressor of paternally expressed imprinted genes and required for silencing sperm-specific genes[40]. We first wanted to see if these factors indeed localize to the endogenous ICRs and analyzed existing mESC ChIP-seq datasets available for SETDB1 (ref. [41]), ZFP57 (ref. [42]), ATF7IP[39] and ZMYM2 (ref. [43]). We observed a strong colocalization of all factors at the endogenous ICRs used in the CRISPR screens (Fig. 5a). By further expanding our analysis to all annotated ICRs, we see that almost all ICRs are co-bound by ATF7IP, ZMYM2, ZFP57 and

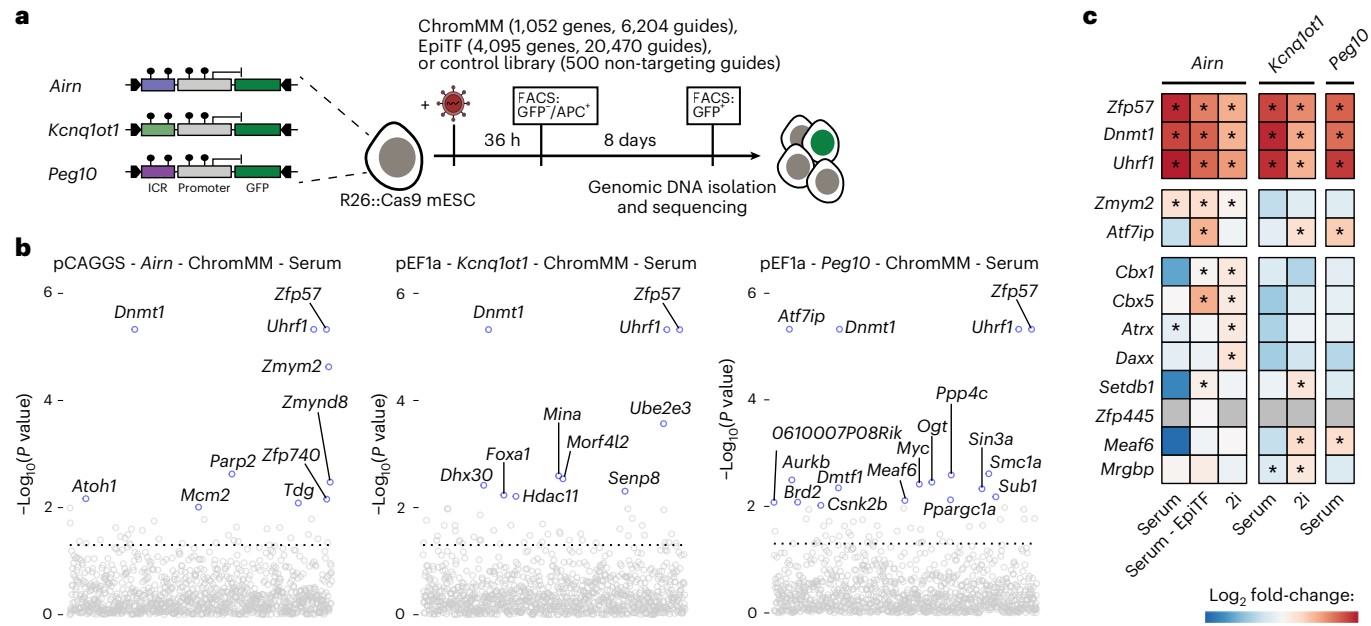

**Fig. 4 | CRISPR screens identify regulators of epigenetic memory at ICRs in mESCs. a**, Experimental overview of targeted CRISPR screens using multiple premethylated ICR reporters. Gating strategy is described in Extended Data Fig. 7a and Methods. **b**, Overview hits from CRISPR screens in three ICR reporter cell lines grown in serum conditions. Blue dots indicate genes with a P value < 0.01 calculated using MAGeCK RRA (robust rank aggregation). Dashed lines indicate the P value threshold at 0.05. **c**, Heatmap showing potential candidates from all CRISPR screens. Color indicates the summarized log fold change across all guides for a given gene, as determined by MAGeCK. Enrichments were calculated combining all replicates for one comparison, using the GFP-enriched fraction against the unsorted pool of cells. Asterisk indicates P < 0.05 using MAGeCK robust rank aggregation. See corresponding panel b and Extended Data Fig. 8b,d. Exact P values can be found in Supplementary Table 1.

SETDB1 (Fig. 5b). Notable exceptions are *MCTS2/H13*, where ATF7IP is absent, and *H19*, which shows a reduced localization of ZMYM2. As a general trend, we observe that ATF7IP and ZMYM2 always colocalize in presence of SETDB1 and ZFP57, suggesting that they localize to ICRs as part of the H3K9me3 machinery. Interestingly, genome-wide analysis of ATF7IP, ZMYM2, ZFP57 and SETDB1 peaks indicates that this colocalization is not always observed outside of ICRs. Although the majority (85%) of the few ATF7IP peaks that we detected overlap with ZFP57 and SETDB1 sites, only 30% of ZMYM2 peaks colocalize with ZFP57 and SETDB1 (Extended Data Fig. 9a). ZMYM2 peaks outside of ZFP57/SETDB1 sites show lower H3K9me3 and DNA methylation compared to peaks overlapping with ZFP57/SETDB1, suggesting that ZMYM2 is involved in multiple regulatory pathways independently of SETDB1 (Extended Data Fig. 9b,c). Regardless of this binding, we see a reduction of ATF7IP and ZMYM2 localization to the *Airn* ICRs in absence of ZFP57 (Extended Data Fig. 9d).

To further interrogate the link between ATF7IP and ZMYM2 at ICRs, we recruited the proximity biotin ligase TurboID[44] to methylated ICRs via a ZFP57-TurboID fusion protein expressed from the RMCE site and performed BioID as previously described[45] (Fig. 5c). As a background control, we generated a cell line expressing just the NLS-TurboID (nTurbo) and included a cell line expressing only the KRAB domain of ZFP57 fused to the TurboID ligase to distinguish between proteins that interact with ZFP57 when not bound to chromatin. Mass-spectrometric detection of enriched proteins included several factors previously associated with ZFP57 (KAP1, CBX3, CBX5 and MORC3). Among them, we detected ATF7IP (Fig. 5c, Extended Data Fig. 9e and Supplementary Table 1), supporting the results obtained from the CRISPR screen and genome-wide analysis. In the case of ZMYM2, we could not detect the protein in the biotinylated fraction or the background sample, suggesting that its enrichment was either below the detection limit or not specifically interacting with ZFP57.

## ATF7IP and ZMYM2 regulate epigenetic memory at endogenous ICRs in mESCs

Next, we wanted to test if absence of these factors that are expressed during early mouse development influences the epigenetic state at endogenous ICRs, and we generated KO mESCs for *Atf7ip* and *Zmym2* using CRISPR-Cas9 (Extended Data Fig. 9f,g). Whole-genome bisulfite sequencing (WGBS) revealed a reduction of DNA methylation at the majority of analyzed ICRs, despite limited loss of methylation genome-wide (Fig. 5d and Extended Data Fig. 10a–c). *Peg13* and *Meg3/Rian* ICRs retained DNA methylation in both KO cell lines, whereas *H13/Mcts* and *Gnas/Nespas* specifically retained methylation in absence of ZMYM2 and *Zrsr1/Commd1* and *H19* in absence of ATF7IP. Loss of ICR methylation was further confirmed by targeted bisulfite sequencing around the binding sites of ATF7IP and ZMYM2 at the *Airn*, *Kcnq1ot1* and *Peg10* ICRs (Extended Data Fig. 10d). Finally, we profiled H3K9me3 in the same KO cell lines and observed loss of H3K9me3 at all ICRs, except for *Peg13* and *Meg3/Rian*, which retained H3K9me3 in both KO lines. In addition, *Zrsr1/Commd1* and *H19* retained H3K9me3 in *Atf7ip* KO cells (Fig. 5e and Extended Data Fig. 10e). The concordant changes in DNA methylation and H3K9me3 at ICRs in the absence of ATF7IP or ZMYM2 indicate that these factors are required for preventing switching of ICRs from methylated to unmethylated states in mESCs.

## Discussion

Here, we set out to study the genetic and epigenetic determinants that allow ICRs to maintain their differential DNA methylation. Toward this, we isolated ICRs from their endogenous chromosomal context and inserted them into a heterologous position in the mESC genome. When integrated unmethylated, the tested ICRs maintained a DNA-methylation-free and euchromatic state, suggesting sequence-specific mechanisms that prevent de novo methylation. This behavior is similar to CpG island promoters, which are protected

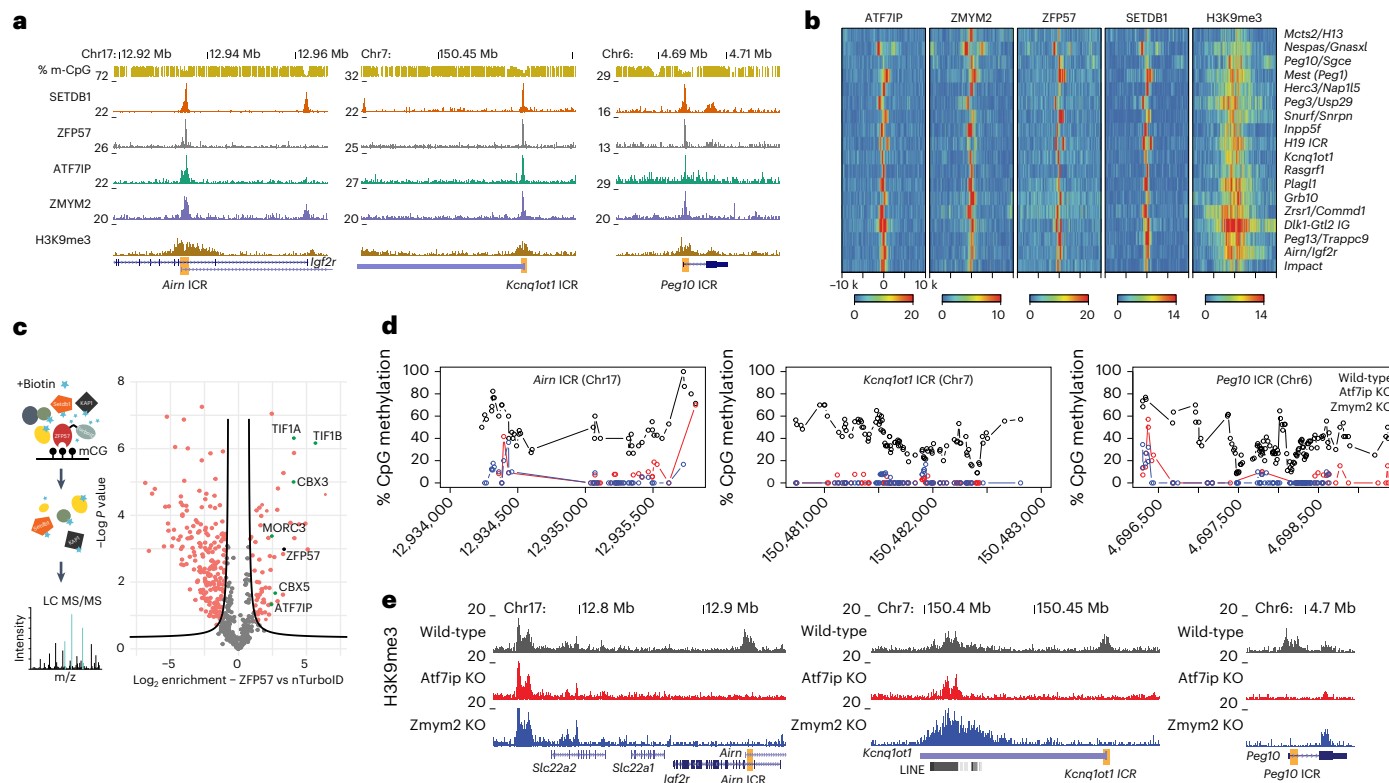

**Fig. 5 | ATF7IP and ZMYM2 colocalize to ICRs and contribute to DNA methylation and H3K9me3 maintenance in mESCs. a**, Genome browser snapshots for the *Airn*, *Kcnq1ot1* and *Peg10* ICRs used in the CRISPR screen experiments. ChIP-seq datasets indicate colocalization of ZFP57, ATF7IP, ZMYM2 and SETDB1 at the ICRs of interest. **b**, Heatmaps summarizing binding of ATF7IP, ZMYM2, ZFP57, SETDB1 and H3K9me3 10 kb (k) upstream and downstream at all annotated ICRs in the mouse genome. Shown are library-normalized reads per 20 bp. **c**, Left: schematic representation of biotin-proximity ligation setup to detect proteins at ZFP57-bound sites comparing ZFP57 fused to TurboID. LC MS/MS: liquid chromatography coupled to tandem mass spectrometry. Right: volcano plot showing enriched proteins and indicating statistically significant

hits from a direct comparison between ZFP57-TurboID and TurboID-NLS (nTurboID). Statistically enriched proteins are indicated (false discovery rate (FDR)-corrected two-tailed *t*-test: FDR = 0.05, Artificial within groups variance (s0) = 1, *n* = 4 technical replicates). **d**, DNA methylation analysis at selected ICRs shows loss of methylation in *Atf7ip*-KO and *Zmym2*-KO cells (see Extended Data Fig. 10c for other ICRs). Shown are methylation values for individual CpGs obtained from WGBS in wild-type, *Zmym2*-KO or *Atf7ip*-KO cells. Genomic position of CpGs is indicated below. **e**, H3K9me3 ChIP-seq indicates loss of H3K9me3 at the selected ICRs (Extended Data Fig. 10e for other ICRs). Shown are reads per 100-bp windows. ICRs in the respective imprinting regions are indicated.

from DNA methylation through elevated CpG density[4,5]. Indeed, based on their CpG density and GC percentage, most ICRs fulfil the definition of CpG islands. In contrast, integration of DNA-methylated ICR sequences to the same site overwrites this default state and leads to stable propagation of DNA methylation with subsequent establishment of heterochromatin marks. Thus, ICRs are autonomous DNA sequence elements that can recapitulate the epigenetic regulatory mechanisms observed at their endogenous position. This finding is in line with previous work indicating that the DNA sequence of ICRs is sufficient to recapitulate the establishment of imprints during mouse development[45–49]. Importantly, this switching between two opposing chromatin states based on DNA methylation was not observed for non-ICR promoters and other DNA sequences of similar size, CpG density or GC content, suggesting that specialized properties of the full-length ICR are required for this 'epigenetic bistability'.

The ectopic ICRs enabled to systematically study the DNA sequences and chromatin regulatory factors required for creating and maintaining epigenetic memory at ICRs in a controlled genomic environment. Through introducing synthetic ICRs with modified DNA sequences, we observe that GC content and CpG density is not sufficient for encoding bistability in mESCs but that additional sequences, such as ZFP57 binding motifs, play an important role in maintaining DNA and H3K9 methylation. This finding is in line with previous work, where mutations of the methylated CpGs of the ZFP57 recognition motif

resulted in loss of methylation maintenance over the entire *Snrpn* ICR[21]. In addition, we show that ZFP57 binding is not only required but also sufficient for the epigenetic memory at the methylated *Airn* ICR state in mESCs. In the case of the unmethylated allele, the same sequence changes result in loss of protection from de novo methylation, suggesting sequence-specific mechanisms that protect from de novo methylation, potentially similar to those observed at regulatory regions of nonimprinted genes[3,5]. Nevertheless, because maintenance of differential *Airn* ICR methylation in MEL cells was independent of DNA sequences outside of CpGs, we suggest a cell-type-specific requirement for sequence-specific factors involved in epigenetic maintenance. In the case of ZFP57, this would be in line with its restricted transcriptional activity to germ cells and during early development[18,29].

Having identified the robust establishment and maintenance of heterochromatin at methylated ICRs, we could generate reporter cell lines that respond to DNA methylation. In contrast to previous strategies that used the *Snrpn* promoter to report changes in methylation at endogenous gene promoters[50,51], our cell lines directly report regulatory changes at the introduced ICRs. We used these reporters to screen for factors required for maintenance of the repressed state. Targeted CRISPR screens identify *Dnmt1*, *Uhrf1* and *Zfp57* as the most relevant genes required to maintain the DNA methylation status at all tested ICRs, confirming the suitability of our setup. In addition, our functional screens identified, and thus validate, additional factors that have been

described to regulate H3K9me3 throughout the genome and to associate with ICRs, including DAXX, ATRX, CBX1 and CBX5 (refs. [14,52,53]).

Among the obtained hits, we identified ATF7IP and ZMYM2 as factors involved in the maintenance of epigenetic repression at ICR reporters. ATF7IP and SETDB1 show functional overlap in the regulation of endogenous retroviral elements, with ATF7IP acting as a cofactor of SETDB1 by stimulating its enzymatic activity, protecting it from proteasomal degradation and facilitating its nuclear localization [35,37,54,55]. Although loss of SETDB1 is lethal in mESCs, the absence of ATF7IP reduces levels of SETDB1, sufficient for viability but insufficient to maintain all repressed sites in the genome [37,56]. The C-terminal fibronectin type-III domain of ATF7IP has been shown to interact with ZMYM2 (also ZFP198), and this interaction was suggested to be important for the silencing of a few germline-specific genes, including the imprinted FKBP6 gene [39,57]. ZMYM2 was also described to interact with H3K9me3-marked chromatin [58,59] and furthermore required for endogenous retroviral element silencing, thereby preventing the transition to two-cell-like cells in mESC culture [43,54]. The role of ZMYM2 in restricting potency is further supported by the fact that ZMYM2 is required for exit from pluripotency [38,60]. We show that ATF7IP and ZMYM2 colocalize together with ZFP57 and SETDB1 at the majority of endogenous ICRs in mESCs and are required for the memory of the epigenetic state at methylated ICRs. This is in line with a publication showing a role of ATF7IP in regulating sperm-specific genes and paternally expressed imprinted genes, including *Peg13* in human parthenogenetic ESCs [40]. Our results indicate that, in mESCs, ATF7IP could play a broader role in regulating all methylated ICRs, independently of the parental origin.

If and how these two factors contribute to maintenance of imprints during zygote formation and early development remains to be tested. In mESCs, their absence resulted in impaired maintenance fidelity and sporadic loss of H3K9me3 at multiple ICRs, independently of their parental origin. We suggest that this destabilizes the repressive feedback loop between DNA methylation and H3K9me3, allowing switching of the ICR to the default unmethylated state. While we observe differences in regulatory activities of ATF7IP and ZMYM2 at some ICRs (for example, *Mcts2/H13*, *Zrsr1/Commd1* and *H19*), it remains to be determined if this is due to a specificity of these factors toward these ICRs. Alternatively, this could reflect stochasticity in ICR switching to an unmethylated state in absence of either factor, which is memorized in clonally derived cells.

## Online content

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

## Methods

### Cell culture

RMCE-competent mESCs (TC-1 (ref. [3]), obtained from A. Dean, National Institutes of Health (NIH)) were cultured on 0.2% gelatin-coated dishes in mESC medium containing DMEM (Invitrogen), 15% fetal calf serum (Invitrogen), 1× non-essential amino acids (Invitrogen), 1× Glutamax (Invitrogen), 0.001% 2-mercaptoethanol (Invitrogen) and titrated leukemia inhibitory factor (made in-house) at 37 °C in 7% $CO_2$. Alternatively, mESCs were cultured in 2i medium containing 50% Neurobasal medium (Invitrogen), DMEM/F12 medium (Invitrogen), 1× non-essential amino acids (Invitrogen), 1× Glutamax (Invitrogen), 0.001% 2-mercaptoethanol (Invitrogen), 1× N2 supplement (Invitrogen), 1× B27 supplement (Invitrogen), titrated leukemia inhibitory factor, 3 μM CHIR99021 (Sigma-Aldrich) and 1 μM PD0325901 (Sigma-Aldrich). Where indicated, L-ascorbic acid (Stemcell Technologies) was added at a concentration of 100 μg ml$^{-1}$ (ref. [26]). Differentiation to neuronal progenitor cells was performed as previously described without feeder cells[61]. For DNMT1 inhibition, GSK-3484862 (MedChemExpress) was added to a final concentration of 10 μM, as previously determined[34]. RMCE-competent MEL cells[30] (obtained from D. Schübeler, FMI Basel) cells were cultured in suspension in DMEM (Invitrogen) supplemented with 10% fetal calf serum (Invitrogen) and 1× Glutamax (Invitrogen). All RMCE-competent cell lines (TC-1 and MEL) were authenticated based on selection and PCR on the RMCE resistance cassette.

### Cell line generation

Targeted cell line integrations in mESCs were obtained through RMCE using either electroporation of $2 \times 10^6$ cells with the Amaxa Nucleofector (Lonza) or Lipofectamine 3000 (Invitrogen) transfections of $2.5 \times 10^4$ cells. All RMCE vectors were cotransfected with a CRE-expressing plasmid at a ratio of 1:0.6 μg, using either a total of 40 μg plasmid for the Amaxa Nucleofector kit or 1 μg for Lipofectamine 3000. Two days after transfection, cells were selected with 3 μM Ganciclovir for more than 8 days. The obtained cell lines were kept as pools and when necessary clonal cell lines were obtained through limited dilution. Pools or clonal cell lines were genotyped using integration site specific PCRs. The parental cell line for all reporter cell lines used in the CRISPR screens contains a stably expressed Cas9 gene from the *Rosa26* locus, obtained by TALEN-mediated integration as previously described[62]. Single-clone KO cell lines were obtained by CRISPR-Cas9 using the px330-hSpCas9 (Addgene, 42230) plasmid together with a pRR-Puro recombination reporter[62]. A total of 1 μg plasmid DNA at a ratio of 1:0.1 of px330 to pRR-Puro was transfected using Lipofectamine 3000. Puromycin selection was started 36 h after transfection for 36–48 h at a concentration of 2 μg ml$^{-1}$. KO cell lines were validated using targeting site-specific PCR. RMCE in MEL cells was performed using Lipofectamine 3000 (Invitrogen), plating $5 \times 10^5$ cells in 6-well plates for suspension cells. A total of 2.5 μg plasmid DNA, using the same ration as described before, was transfected according to the manufacturer's instruction. After 48 h, cells were transferred into T75 flasks, and cells that underwent recombination were selected with 5 μM Ganciclovir containing media for more than 8 days.

### Reporter cell line generation

A backbone containing two inverted *loxP* sites[3] was used to clone several empty reporter vectors containing a 60-bp universal entry site with a central EcoRV restriction site, followed by a promoter (pCAGGS, hPGK and Ef1alpha) that drives an eGFP or mScarlet gene for the ChroMM and EpiTF screens, respectively, followed by a downstream BGH-poly(A) and a WPRE sequence. Individual ICR or control sequences were amplified from genomic DNA (Supplementary Table 1). Gibson assembly was performed according to the NEB Gibson Assembly Master Mix protocol. In vitro methylation was performed with up to 40 μg plasmid DNA using the NEB M.SssI methyltransferase in two consecutive reactions of at least 4 h with 600 μM SAM (NEB, B9003S) and 1.5 U M.SssI (NEB, M0226L) per microgram DNA. Complete methylation of plasmids was confirmed by using the CpG methylation sensitive restriction enzyme HpaII (NEB) and a methylation insensitive control reaction with MspI (NEB). Cell lines were generated as described before. Individual clones were genotyped using PCR with primers spanning the *loxP* sites. Methylation of the integrated reporter construct was validated on selected clones.

### Flow cytometry and fluorescence-activated cell sorting (FACS)

Flow cytometry data acquisition was performed on a BD FACSCanto II or a BD LSR Fortessa cell analyzer. FACS was performed with a BD FACSAria III cell sorter. Data analysis was done with FlowJo (version 10.7) or BD FACSDiva (9.1.2). All samples were gated for single cells, using forward scatter area (FSC-A) versus side scatter area (SSC-A), followed by FSC-A versus forward scatter height (FSC-H). GFP-negative and positive populations were quantified using GFP-negative wild-type cells as a reference. For cell surface marker staining, a uniform cell suspension was prepared by trypsinization and filtering through a 40-μm cell strainer (BD Bioscience). Cells were stained with an allophycocyanin (APC)-conjugated CD90.1 antibody (Invitrogen, 17-0900-82) for 30 min at 4 °C with a saturated antibody concentration (1 μl per 15 million cells).

### In silico sequence analysis using BPnet

BPnet[28] (version 0.0.23) was used to determine sequence motif and context of ZPF57 binding in mESCs. ZFP57 ChIP-seq data and corresponding input files[42] were aligned to the mouse genome (NCBI Build 37 mm9, July 2007) using bowtie2 (version 2.3.5.1) after removal of adapters using trimgalore (version 0.6.6). Aligned reads were filtered for PCR duplicates using Picard (version 2.23.9), and only reads with a mapping quality (MAPQ) > 40 were kept for further analysis. All replicates were merged before peak calling using MACS2 (version 2.1.1.20160309) with the following parameters: callpeak -g mm –keep-dup all -q 0.05– call-summits. Reads mapped to the positive and negative strand of the merged datasets were split into individual files and trimmed to the 5′ base as input tracks for BPnet. A model was trained with the default bpnet9 architecture (https://github.com/kundajelab/bpnet), using chromosomes 1, 8 and 9 as test set, and peaks on chromosomes 2, 3 and 4 as validation sets. Peaks on chromosomes X and Y were excluded from model training. After calculation of the contribution scores with BPnet's DeepLIFT method, motifs were determined using BPnet's TF-MoDISco method. To determine contribution scores on the *Airn* and shuffled *Airn* sequences, the input DNA was one-hot encoded before subjecting them to the trained model to generate ZFP57 binding predictions. For the walking mutations, 10 nt of the shuffled sequence was swapped with the original *Airn* sequence and shifted by 1 bp per prediction.

### Bisulfite PCR and sequencing

Up to 2 μg genomic DNA, or the total amount to eluted material from ChIP, was used for bisulfite conversion using the EpiTect Bisulfite Kit (Qiagen). Bisulfite PCR was carried out using the PhusionU polymerase (Thermo Fisher Scientific) with the primers indicated in Supplementary Table 1 using the following conditions: initial denaturation at 95 °C for 5 min, followed by 45 cycles of 1 min at 95 °C, 1 min at 50–60 °C (dependent on the primer pair) and 1 min at 72 °C, followed by 5 min of final extension at 72 °C. Amplicons were cloned into the CloneJET vector (Thermo Fisher Scientific), sequenced by Sanger sequencing and analyzed using QUMA[63].

### Targeted bisulfite sequencing

Targeted bisulfite sequencing libraries were made from equimolar pooled bisulfite PCR fragments. Two independent PCR reactions were run per target with annealing temperatures at 50 °C and 58 °C to mitigate amplification bias. Indexed libraries were prepared using

the NEBNext Ultra II kit (NEB) starting from 10 ng pooled amplicons according to the manufacturer's protocol. Sequencing was done on an Illumina NovaSeq6000 machine with 150-bp paired-end reads. Fastq files were trimmed using trim_galore (version 0.6.6) and alignment was performed with Bismark (version 0.23.0) with the parameter non_directional. CpG methylation was extracted using the Bismark methylation extractor and average CpG methylation was calculated in R, excluding CpGs that were covered less than 500 times.

### Whole-genome bisulfite library preparation and sequencing
WGBS of *Atf7ip* and *Zmym2* KO mESCs was performed as described previously[64]. In short, 10 μg genomic DNA was sonicated to a length of approximately 400–500 bp. For each sample, 2 μg sheared genomic DNA was mixed with 10 ng equimolar pooled sonicated methylated phage T7 and unmethylated phage Lambda DNA. Adapter-ligation was carried out with the NEBNext Ultra II kit (NEB E7645L) using methylated adaptors (NEB, E7535S), before bisulfite conversion using the Qiagen Epitect bisulfite conversion kit, according to the manufacturer's instructions. After conversion, libraries were amplified for 10 cycles using the Pfu TurboCx Hotstart DNA polymerase (Agilent) and the NEB dual index primers (NEB, E7600S). PCR reactions were run with the following parameters: 95 °C for 2 min, 98 °C for 30 s, followed by 10 cycles of 98 °C for 15 s, 65 °C for 30 s and 72 °C for 3 min, ending with 5 min at 72 °C. The PCR reactions were cleaned up using 1.2× AMPure XP beads (Beckman Coulter) and eluted in 20 μl EB buffer (Qiagen). Library quality was checked on an Agilent TapeStation and sequencing was done on an Illumina NovaSeq 6000 machine.

### ChIP
Approximately $15 \times 10^6$ cells were harvested per IP and fixed with 1% methanol-free formaldehyde for 8 min. Crosslinking was quenched by adding glycine to a final concentration of 0.125 nM and incubated for 10 min at 4 °C on ice. Cells were pelleted at 600 × g for 5 min, washed with cold PBS and incubated for 10 min on ice in a buffer containing 10 mM EDTA, 10 mM Tris pH 8, 0.5 mM EGTA and 0.25% Triton X-100. After centrifugation, cells were incubated in 1 mM EDTA, 10 mM Tris pH 8, 0.5 mM EGTA and 200 mM NaCl for 10 min on ice. Chromatin was extracted in a high-salt buffer containing 50 mM HEPES, pH 7.5, 1 mM EDTA, 1% Triton X-100, 0.1% deoxycholate, 0.2% SDS and 500 mM NaCl for 2 h at 4 °C, and chromatin was sheared using a Bioruptor Pico sonicator (Diagenode). Then, 100 μg of chromatin was used per IP reaction with 30 μl pre-blocked magnetic Protein A beads (Invitrogen). Beads were blocked with 1 mg BSA and 100 ng yeast tRNA (Sigma) in TE buffer containing proteinase inhibitor mix (Roche) before use. Prior to the IP, chromatin was precleared with 20 μl blocked beads for 1 h at 4 °C. Next, 5% of input material was kept at −20 °C and decrosslinked along the IP material. Then, 5 μg antibody was used per IP for overnight incubation at 4 °C. The next day, 30 μl blocked beads was added to the chromatin and incubated for 4 h at 4 °C. Beads were separated on a magnet and washed twice for 8 min with high-salt buffer, one time with 50 mM LiCl, 0.5% NP-40, 0.5% deoxycholate, 1 mM EDTA and 10 mM Tris, pH 8. After two additional washes with TE for 8 min, chromatin was eluted after 30 min incubation at 37 °C with 60 μg RNaseA (Roche) in 1% SDS and 100 mM NaHCO₃, followed by 3-h incubation adding 10 mM EDTA, 40 mM Tris, pH 8, and 60 μg Proteinase K (Roche). Final decrosslinking was done overnight at 65 °C. Eluted material was cleaned up using phenol chloroform extraction and quantified using a Qubit 2.0 fluorometer (Thermo Fisher Scientific). The following antibodies were used for ChIP: H3K9me3 (Abcam, ab8898, 5 μg per IP), H3K4me2 (Diagenode, C15410035, 5 μg per IP), H3K4me3 (Abcam, ab8580, 5 μg per IP), ATF7IP (Bethyl, A300-169A, 5 μg per IP) and ZMYM2 (Bethyl, A301-711A, 10 μg per IP).

### ChIP-qPCR
qPCR reactions were run as technical duplicates on a Rotor Gene Q machine (Roche) using the KAPA SYBR Fast universal qPCR kit (Sigma) in 10-μl reactions with 1 μl eluted DNA for the IP material or 1 μl of a 1:10 dilution of the input material. Delta Ct values were calculated over input, followed by delta Ct and fold change over an intergenic region. Primers are listed in Supplementary Table 1. Corresponding plots were generated with Prism (5.0a).

### ChIP-seq library preparation
ChIP-seq libraries were prepared using the NEBNext ChIP-seq Library Prep Master Mix set for Illumina (NEB, E6240) or NEBNext Ultra II Kit, following the manufacturer's protocol. Final libraries were visualized and quantified on a 2200 TapeStation System (Agilent) and pooled with equal molar ratios before sequencing on an Illumina NovaSeq6000 machine with 150-bp paired-end reads.

### Genome-wide datasets and analysis
Published mESC genome-wide datasets were obtained from GEO (WGBS[65]; H3K9me3, H3K4me3, H3K36me3 and H3K27me3 (ref. [66]), DNase-seq and RNA-seq[67], SETDB1 (ref. [41]), ZFP57 (ref. [42]), ZMYM2 (ref. [43]) and ATF7IP[39]. Sequencing reads from published datasets and ChIP-seq reads generated in this study were filtered for low-quality reads as well as adaptor sequences using trimgalore (version 0.6.6) and mapped to the mouse genome (NCBI Build 37 mm9, July 2007). Mapping of H3K9me3, H3K4me3, H3K36me3, H3K27me3, DNase-seq and RNA-seq was done with QuasR (1.30.0) in R with standard qAlign() settings. Wig tracks were obtained with QuasR qExportWig() command and visualized using the UCSC genome browser (https://genome.ucsc. edu). Mapping of WGBS data was done with QuasR using qAlign() with following settings: genome = 'BSgenome.Mmusculus.UCSC.mm9', aligner = 'Rbowtie' and bisulfite = 'dir'. CpG methylation calls were extracted using qMeth() and filtered to contain only CpGs covered at least 10×. ChIP-seq peak coordinates were obtained using MACS2 (version 2.1.1.20160309) with the following parameters: callpeak -g mm– keep-dup all -q 0.05 –call-summits. Coordinates were imported into R as GenomicRanges objects and peaks larger than 1 kb were removed from further analysis. Overlaps between peaks were calculated using the findOverlaps() function in R, with maxgap=1000 L. Heatmaps over ICRs and peak regions were generated with genomation() in R using the ScoreMatrixList() and multiHeatMatrix() functions.

### CRISPR libraries and screens
The ChromMM and EpiTFs library was constructed as a subpool of the Vienna sgRNA library as described previously[68]. Lentivirus was produced in HEK293T (obtained from G. Schwank, University of Zurich) cells as described[69]. Nonconcentrated virus was titrated with different amounts following the same transduction procedure used for the actual CRISPR screens. Transduction was performed with $1.25 \times 10^6$ cells seeded in gelatin-coated 6-well plates in embryonic stem cell medium containing 8 μg ml⁻¹ polybrene (Merck), spinning for 60 min at 500 × g at 37 °C. After centrifugation, cells were incubated for 12 h at 37 °C, before transferring them on multiple 15-cm plates and culturing them for another 24 hours. After 36 h, transduced cells were selected using FACS. For the ChromMM library cells were stained against the CD90.1 cell surface marker using an APC-conjugated antibody (Invitrogen, 17-0900-82, 1 μl per 15 million cells), gating on APC-positive and GFP-negative single cells. For the EpiTF library, cells were gated on GFP-positive and mScarlet negative single cells. After the sort, cells were seeded sparsely on multiple 15-cm dishes and only passaged once after 4 days to avoid bottlenecks in the library representation. On day 10 after transduction, GFP-positive cells were sorted and further processed for genomic DNA extraction using the DNeasy Blood and Tissue kit (Qiagen). All screens were performed with at least 30 million cells and a low multiplicity of infection between 0.1 and 0.2, yielding a total cell number of at least 3 million cells and a guide representation of at least 450× per guide after the first sort. For the final sort, the same number of initially transduced cells were used for the sort and kept as

the reference pool. The screen with the EpiTF library was performed as described above, however 90 million cells were used for transduction at an multiplicity of infection of 0.2. Initially, all screens were performed as technical duplicates or triplicates with individual reporter clones and later repeated once more as independent experiments. To score essential and growth-restricting genes, the pooled cells at indicated days were compared to the initial plasmid library.

## Library preparation for CRISPR screen

Library preparation was done for the entire amount of extracted genomic DNA in two consecutive PCR amplification steps. In the first PCR, the integrated guide sequences were amplified using the Herculase II Fusion DNA polymerase (Agilent) according to the manufacturer's instruction with a maximum of 500 ng DNA input per 50 µl reaction using library specific primers with 3' adapter sequences for barcoding (Supplementary Table 1). The PCR mix contained 1.5% DMSO and had a final concentration of 3 nM of $MgCl_2$. Amplified products were first purified using the MinElute Gel extraction kit (Qiagen) and potential primer dimers were removed using AmpureXP beads (Beckmann) at a ratio of 0.7× volume. Sample specific barcoding was done in a second PCR using NEBNext Multiplex Oligos (NEB) and the NEBNext Q5 Hot Start HiFi PCR Master mix (NEB) according to the manufacturer's manual with 10% of the eluted product from the first PCR and 7 amplification cycles. For the EpiTF library, barcoding was done with the i5 primers from the NEBNext Multiplex Oligo kit (NEB) and a custom primer that carries the P7 sequence (Supplementary Table 1). Sequencing was done on an Illumina NovaSeq6000 or a MiSeq machine, specifying a 10-bp index read 1 for the EpiTF library.

## CRISPR screen analysis

Demultiplexing was performed using the standard pipeline of Illumina. For the EpiTF library, demultiplexing was only performed on the i5 index, as the i7 index contains the UMI[68]. Fastq files were trimmed to only include the guide RNA sequence using cutadapt (version 3.10) specifying -g 5'-TAGCTCTTAAAC...GGTGTTTCGTC-3' for the linked adapter sequences in the lentivirus backbone for the ChromMM library or -g aaacaccg…gtttaaga for the EpiTF library. Alignment was done using bowtie2 (version 2.3.5.1) against a reference genome built from the sgRNA sequences, specifying the following alignment parameters: -k 1 –very-sensitive. BAM files were converted into bed files using bedtools (version 2.27.1) bamtobed function. Bed files were imported into R to create a count matrix for MAGeCK (version 0.5.9.2). For the final analysis, counts from technical replicates as well as different GFP high and GFP low bins were aggregated. MAGeCK was run with –norm-method set to total and run against the unsorted pool as the control sample, specifying the independent replicates.

## CRISPR screen validation

Single-guide validation was done with one guide RNA that was included in the library and one independently designed guide with high on-target and low off-target activity, as described in ref. [70] (Supplementary Table 1). Guides were cloned into the px459 backbone (Addgene, 62988) which allows for puromycin selection. For this, 1,000 cells were seeded in gelatin-coated wells of a 96-well plate 1 day before transfection. Next, 100 ng plasmid DNA was transfected per well as technical replicates using Lipofectamine 3000 (Invitrogen). After 36 h, transfected cells were selected for 36–48 h with 2 µg ml⁻¹ puromycin using untransfected cells as a control. Reactivation of the reporter was evaluated 12 days after transfection using flow cytometry, and GFP reactivation was quantified over cells transfected with nontargeting control guides.

## Western blot

For western blotting, 20–35 µg protein was separated on 6% or 10% polyacrylamide gels and transferred on polyvinylidene fluoride membranes using the TransBlot Turbo system (Bio-Rad). For antibody-based staining, the membrane was washed once with TBS-T (10 mM Tris, pH 8.0, 150 mM NaCl and 0.1% Tween-20), blocked with 5% non-fat dry milk in TBS-T and stained with primary antibodies against ATF7IP (Bethyl, A300-169A), ZMYM2 (Bethyl, A301-711A-M), or LAMIN B1 (Santa Cruz Biotechnology, 374015) at 4 °C overnight. Next day, membranes were washed three times with TBS-T for 10 min before incubation with species-specific horseradish peroxidase-conjugated secondary antibodies for 1 h at room temperature. After additional three washes with TBS-T for 10 min each, signal was detected using the Amersham ECL Western blotting detection reagent (GE Healthcare Life Sciences; RPN2109) and exposure on Amersham Hyperfilm ECL (GE Healthcare Life Sciences; 28906836) in a darkroom.

## Cell line generation for proximity ligation experiments

Cell lines were generated as described in Villasenor et al.[45]. The coding sequence of the BioID2 enzyme of the original entry vector was exchanged for the coding sequence of the TurboID enzyme[44]. Cells were either transfected with the entry vector, containing only the TurboID with a nuclear localization sequence, the full-length mouse ZFP57 cDNA sequence cloned upstream of the TurboID, or the KRAB domain of ZFP57 as annotated on UniProt. All cells were validated using western blot of cells incubated with 50 µM biotin (Sigma-Aldrich) for 12 h as previously described with minor adjustments to accommodate the biotin detection. In short, membranes were blocked in 5% BSA in TBS containing 0.1% Triton X-100 for 1 h and stained with streptavidin-horseradish peroxidase (1:20,000) in TBS containing 0.1% Triton X-100 overnight at 4 °C. The membrane was washed twice with TBS containing 0.3% Triton X-100, twice with TBS containing 0.3% Triton X-100 and additional 500 mM NaCl, before one final wash with TBS containing 0.3% Triton X-100 for 10 min at room temperature each.

## Proximity ligation using TurboID

TurboID samples were prepared as described in Villaseñor et al.[45]. In brief, cells were grown as quadruplicates on 15-cm plates, incubated with 50 µM biotin (Sigma-Aldrich) for 12 h upon 70% confluency and harvested with trypsin. In the following, samples were handled at 4 °C or on ice. Cell pellets were swelled in 5× volume of nuclear extraction buffer 1 (NEB1; 10 mM HEPES, pH 7.5, 10 mM KCl, 1 mM EDTA, 1.5 mM $MgCl_2$, 1 mM dithiothreitol (DTT), 1× EDTA-free complete protease inhibitor cocktail (PIC; Roche) for 10 min, before spinning at 2,000 × g for 10 min. Cells were homogenized using a loose Dounce pistil in 2× volumes of NEB1. Nuclei were collected by centrifugation at 2,000 × g for 10 min, resuspended in 1× volume nuclear extraction buffer 2 with 450 mM NaCl (NEB2; 20 mM HEPES, pH 7.5, 0.2 mM EDTA, 1.5 mM $MgCl_2$, 20% glycerol, 1 mM DTT and 1× PIC) and homogenized 10 more times with a tight Dounce pistil, followed by an incubation for 1 h with overhead rotation. Debris was removed by centrifugation at 2,000 × g for 10 min before adjusting the salt concentration of the supernatant to 150 mM NaCl with 2× volumes of NEB2 and adjusting the final NP40 concentration to 0.3%. Protein extracts were quantified using the Qubit Protein Assay Kit (Thermo Fisher Scientific, Q33211) and equal amounts of protein extracts were used per IP. For each IP, 40 µl of Streptavidin M-280 Dynabeads (Thermo Fisher Scientific), pre-blocked in IP buffer (IPB; NEB2, 150 mM NaCl, 0.3% NP40, 1 mM DTT, 1× PIC) containing 1% cold fish gelatin, were added to the extracts, and incubated at 4 °C overnight while rotating. Next, beads were washed twice with 2% SDS in TE containing 1 mM DTT and 1× PIC for 10 min rotating at RT, followed by one 10 min wash with a high salt buffer (50 mM HEPES, pH 7.5, 1 mM EDTA, 1% Triton X-100, 0.1% deoxycholate, 0.1% SDS, 500 mM NaCl, 1 mM DTT, 1× PIC), one wash with DOC buffer (50 mM LiCl, 10 mM Tris, pH 8.0, 0.5% NP40, 0.5% deoxycholate, 1 mM EDTA, 1 mM DTT and 1× PIC) and twice with TE buffer containing 1 mM DTT, 1× PIC. After the washes, beads were pre-digested with 5 µg ml⁻¹

trypsin (Promega; V5111) in 40 µl digestion buffer (1 M urea in 50 mM Tris, pH 8.0, 1 mM Tris-(2-carboxyethyl)-phosphine) for 2.5 h at 26 °C and shaking at 600 rpm. The supernatant was further reduced with 2 mM Tris-(2-carboxyethyl)-phosphin for 45 min at room temperature, alkylated with 10 mM chloroacetamide for 30 min at room temperature and protected from light. For the final digest, the protein solution was incubated with additional 0.5 µg trypsin overnight at 37 °C. The next day, the digested samples were prepared for loading on C18 StageT-ips by addition of trifluoracetic acid (TFA) to a final concentration of 0.5% and acetonitrile (ACN) to a final concentration of 3%. In-house produced C18-StageTips (Functional Genomics Center Zurich) were humidified with 100% methanol, cleaned twice with the elution solution (60% ACN, 0.1% TFA) and prepared for loading by washing twice with 3% ACN and 0.1% TFA. After loading of the peptide solution, samples were centrifuged and the supernatant was loaded on more time, before washing twice with 3% ACN and 0.1% TFA. Finally, peptides were eluted twice with the elution solution, shock frozen in liquid nitrogen, dried in a speed vacuum centrifuged and reconstituted in 3% ACN, 0.1% formic acid, containing internal retention time standard peptides (iRTs, Biognosys). Samples were run on an Easy-nLC 1000 HPLC system coupled to an Orbitrap Fusion mass spectrometer (Thermo Fisher Scientific) with block randomized samples

### Protein identification and label-free protein quantification

MaxQuant (version 1.5.3.30) was used for protein identification and label-free quantification[71] based on the mouse reference proteome (UniProtKB/Swiss-Prot and UniProtKB/TrEMBL) version 2018_12 combined with manually annotated contaminant proteins, with a protein and peptide FDR values set to 1%. Perseus was used for statistical analysis as described previously[72]. For this, only proteins were kept that were identified in three out of four samples per group. Missing values were imputed from a 1.8 standard deviations left-shifted Gaussian distribution with a width of 0.3. A $t$-test was used to identify potential interactors using an FDR threshold of < 0.05 and an S0 value of 1. Data were visualized using R (version 4.0.3).

### Reporting summary

Further information on research design is available in the Nature Research Reporting Summary linked to this article.

### Data availability

Sequencing data have been deposited to NCBI GEO under the following accession number GSE176461; The mass spectrometry proteomic data have been deposited to the ProteomeXchange Consortium via the PRIDE partner repository with the dataset identifier PXD034918. Source data are provided with this paper.

### Code availability

All the analyses were performed using previously published or developed tools. No custom code was developed or used.

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

### Acknowledgements

We thank A. Dean (NIH/NIDDK) for kindly providing the TC-1 mESC line and M. Buehler (FMI Basel) and C. Földy (University of Zurich) for sharing plasmids. Furthermore, we thank G. Schwank and group (University of Zurich) for their help in establishing the CRISPR-screen protocol, the Functional Genomics Centre Zurich (FGCZ), S3IT Zurich, and Cytometry Facility at University of Zurich for their support. We thank P.-A. Defossez (Paris Diderot University) and members of the Baubec group for their input and for their constructive criticism. T.B. acknowledges support from the University of Zurich and Utrecht University, Swiss National Science Foundation (183722, 180354 and 190378), the European Research Council (865094 - ChromatinLEGO - ERC-2019-COG) and the EMBO Young Investigator program. N.S. was supported by postdoctoral fellowships from EMBO and University of Zurich and is an SNSF Ambizione grant fellow (186012). R.V. acknowledges support from the University of Zurich (UZH postdoctoral grant FK-21-060). D.S. acknowledges support from the Novartis Research Foundation, the European Research Council under the European Union's (EU) Horizon 2020 research and innovation program grant agreement (667951-ReadMe and 884664-DNAccess) and the Swiss National Science Foundation (310030B_176394). A.R.K. acknowledges support from the European Molecular Biology Laboratory, Deutsche Forschungsgemeinschaft (KR 5247/1-1) and the Swiss National Fund Ambizione (grant PZOOP3_161493).

### Author contributions

S.B. and T.B. designed the study. S.B., N.S., I.K., R.V., I.S., S.D., F.L., N.d.W., X.B. and T.B. performed experiments. J.J., U.E., J.Z. and U.E. designed and generated targeted CRISPR libraries. A.K. and D.S. provided cell lines and datasets used in this study. S.B. and T.B. wrote the manuscript, with input from all authors.

### Competing interests

The authors declare no competing interests.

### Additional information

**Extended data** is available for this paper at https://doi.org/10.1038/s41588-022-01210-z.

**Correspondence and requests for materials** should be addressed to Tuncay Baubec.

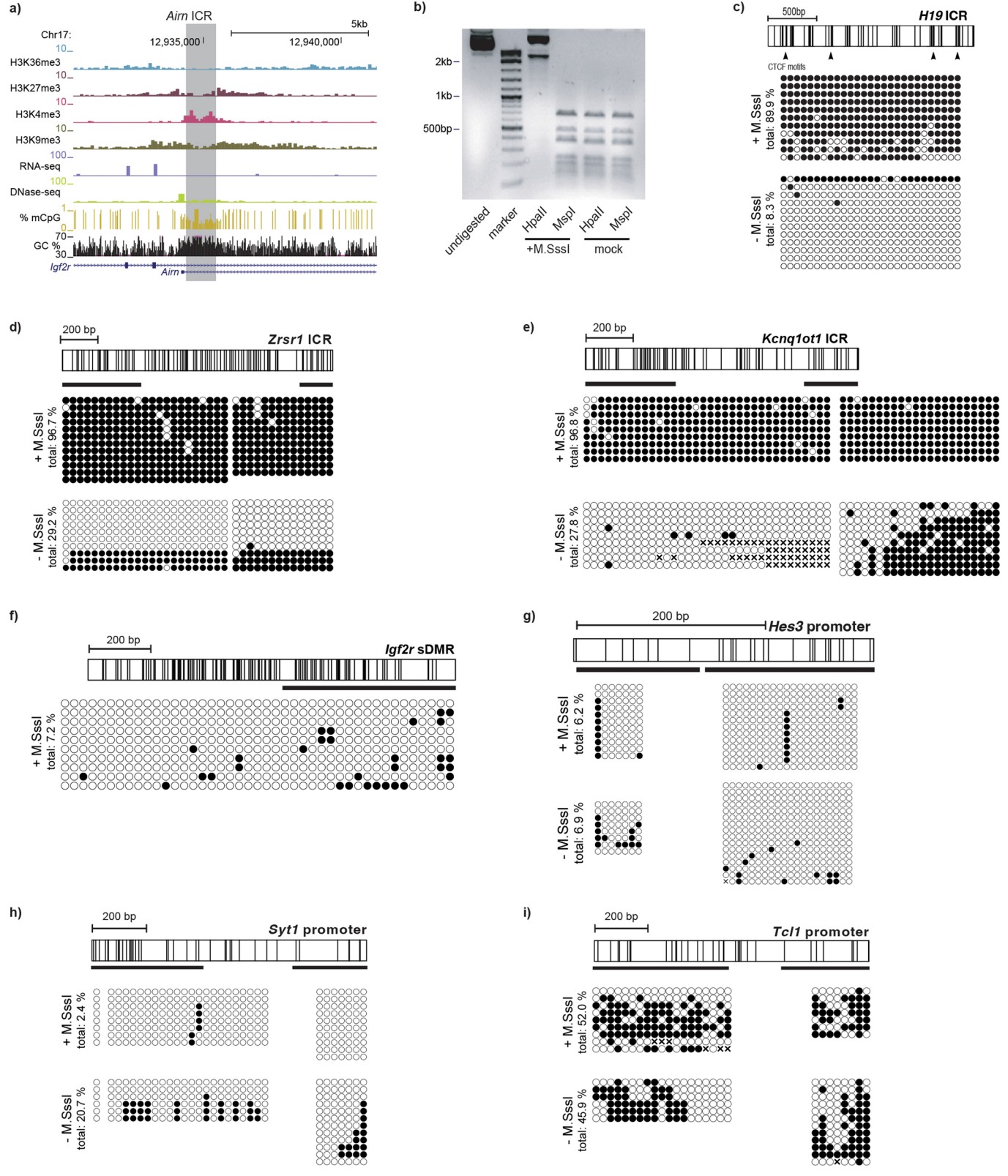

**Extended Data Fig. 1 | See next page for caption.**

**Extended Data Fig. 1 | ICR sequences maintain pre-established methylation levels at an ectopic integration site. a**) Genome browser snapshot for the for the *Airn* ICR locus in mouse embryonic stem cells. Shown are ChIP-seq tracks for chromatin modifications, RNA-seq data for transcriptional activity, DHS-seq for chromatin accessibility, WGBS data for DNA methylation and local GC density in percent. The ICR sequence used for experiments is highlighted. **b**) Representative agarose gel for restriction digest of *in vitro* methylated and unmethylated plasmids prior to transfection. *HpaII* and *MspI* share the same recognition site, however *HpaII* is blocked by CpG methylation. Experiment was repeated at least twice prior to RMCE integration. Molecular size markers are indicated. **c**) Schematic overview of the ectopic integrated DNA fragment for *H19* with vertical lines illustrating individual CpG sites (top) and single molecule measurements of DNA methylation of the pre-methylated and unmethylated sequences using bisulfite PCR (bottom). **d-e**) same as in (c): for *Zrsr1* (d) and *Kcnq1ot1* (e). **f**) Schematic overview of the ectopic integrated DNA fragment for the *Igf2r* gametic DMR with vertical lines illustrating individual CpG sites (top) and single molecule measurements of DNA methylation of the pre-methylated and unmethylated sequences using bisulfite PCR (bottom). **g-i**) same as in f: *Hes3* (g), *Syt1* (h), and *Tcl1* (i). Unmethylated (- M.SssI) bisulphite data for g-i was obtained from Lienert et al.[3] and shown for comparison.

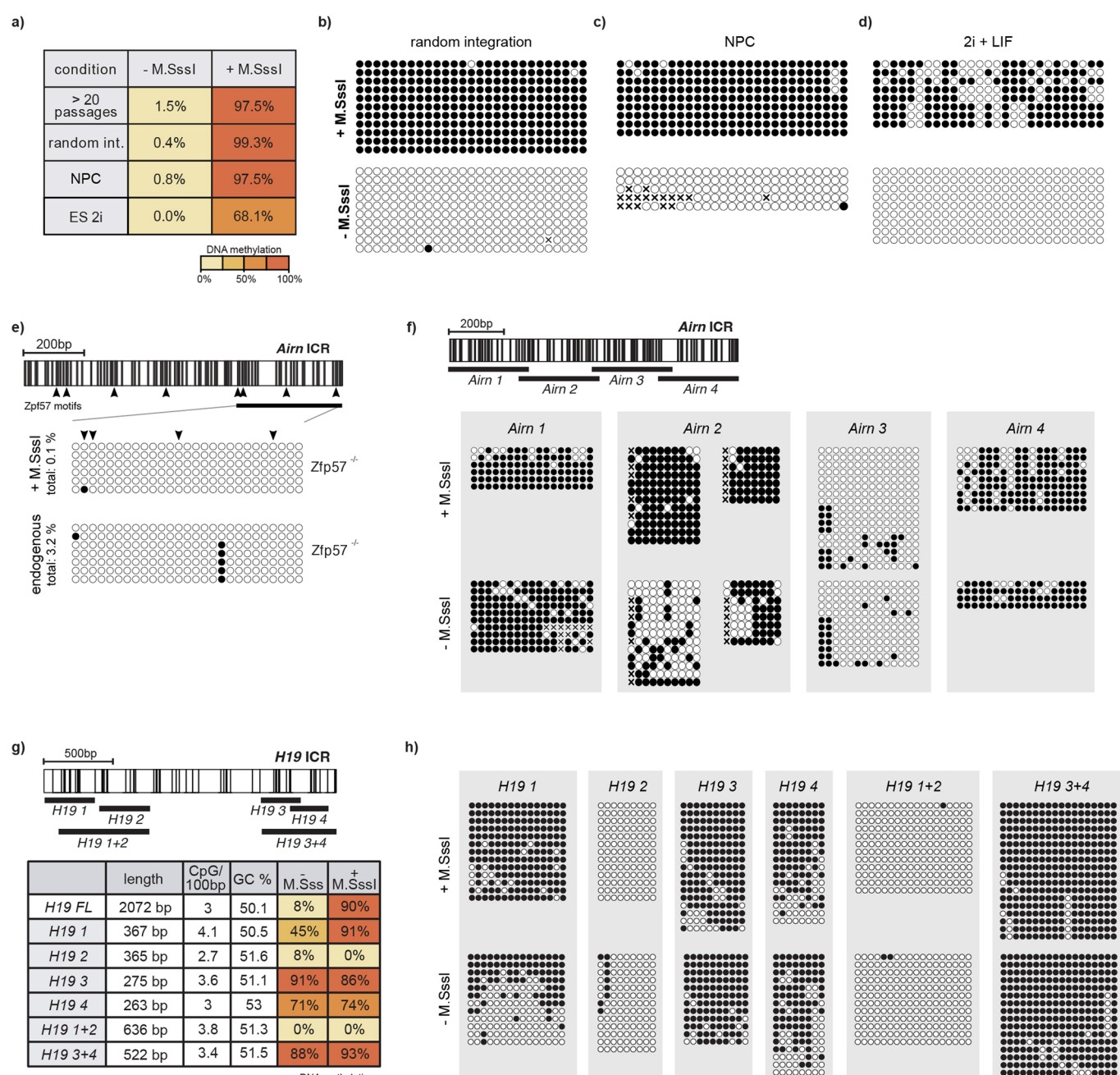

**Extended Data Fig. 2 | Stable methylation maintenance at ectopic ICRs.**
**a)** Summary table for the methylation analysis of the *Airn* ICR after long term culture (over 20 consecutive passages), random integration to a different genomic site, during differentiation to neuronal progenitor cells (NPCs), or after culturing in 2i for 10 days. **b-d)** Single molecule measurements from bisulfite PCR corresponding to data summarized in a. **e)** Methylation analysis for the ectopic and endogenous *Airn* ICR after CRISPR-mediated *Zfp57* KO. Triangles

indicate CpGs within ZFP57 motifs. The region analysed by bsPCR is indicated. **f)** Schematic overview of *Airn* ICR fragments tested in the study (top) and single-molecule bisulphite PCR results from the individual fragments (below) **g)** Summary table for the methylation analysis of the *H19* ICR fragments, including size, CpG density and GC content information. **h)** Single molecule measurements from bisulfite PCR corresponding to data summarized in g.

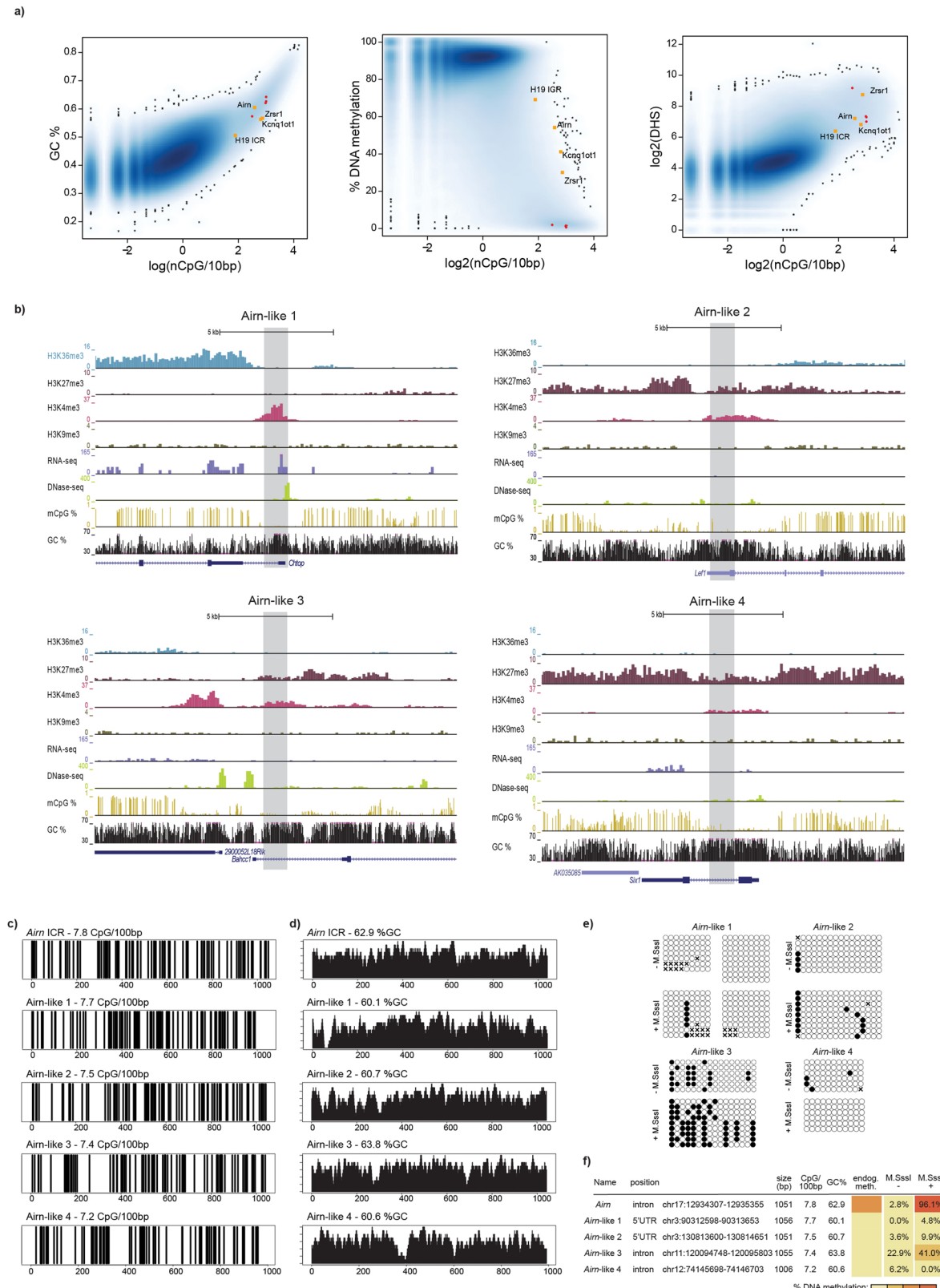

**Extended Data Fig. 3 | Identification of elements with similar sequence characteristics to the *Airn* ICR. a)** Analysis of sequence characteristics for genome-wide 1 kb windows. Yellow dots indicate ICR sequences used in this study. Red dots indicate *Airn*-like fragments. **b)** Genome browser snapshots for all four *Airn*-like sequences. Highlighted boxes indicate the DNA sequences used in the RMCE experiments. **c)** Sequence characteristics of selected *Airn*-like sequences compared to the *Airn* ICR. Vertical lines correspond to individual CpG sites within the sequence. **d)** GC percentage of selected *Airn*-like sequences. **e)** Single molecule representation of data summarized in f. **f)** Tabular summary of methylation analysis for all *Airn*-like sequences, including the *Airn* ICR for comparison.

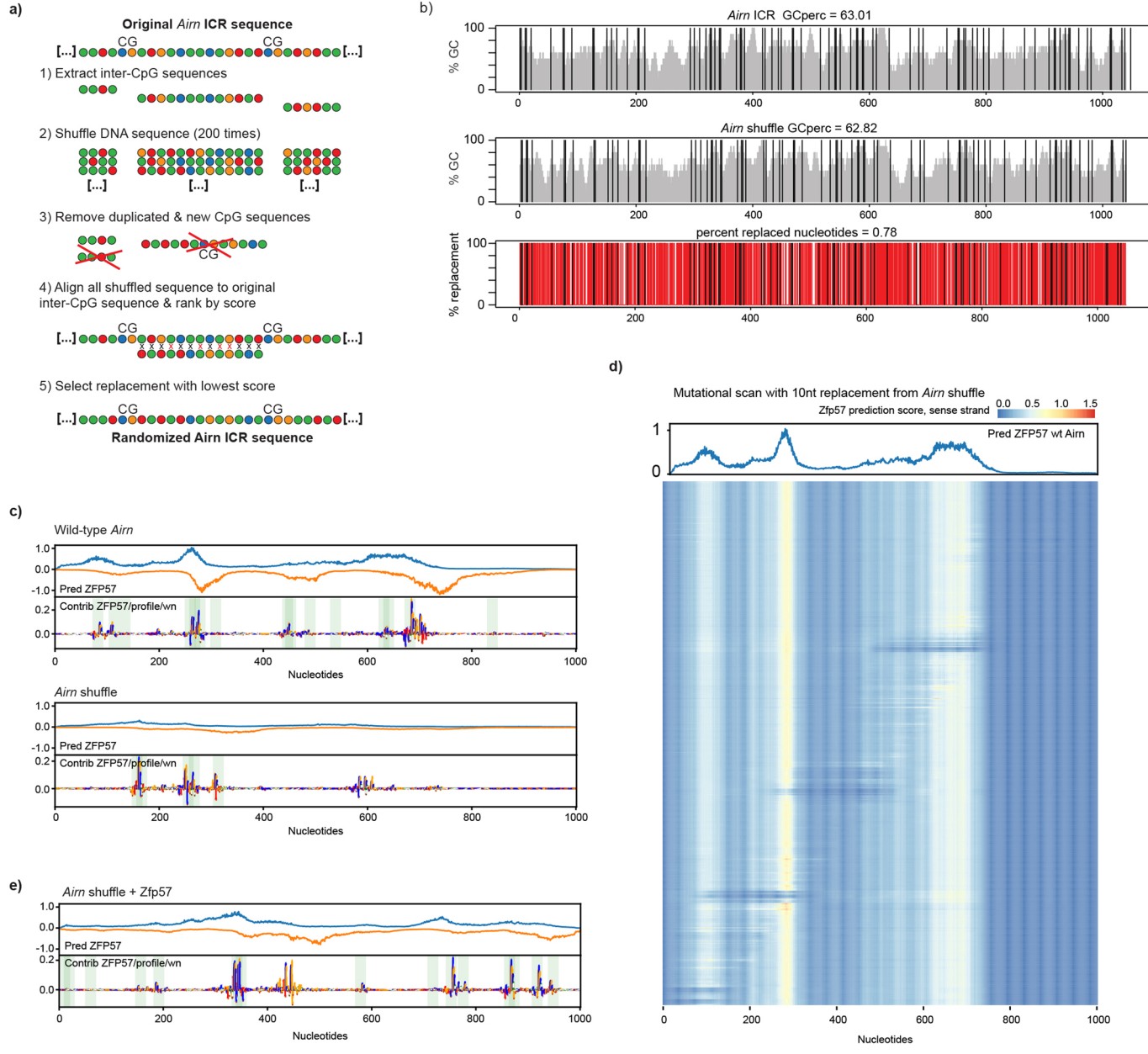

**Extended Data Fig. 4 | Methylation maintenance of *Airn* ICR is independent of CpG positioning and GC content but requires ZFP57 motifs. a)** Workflow indicating in silico sequence shuffling for the *Airn* ICR. **b)** Sequence characteristic of most-divergent shuffled sequence (*Airn* shuffled) compared to the original *Airn* sequence. **c)** BPNet evaluation of ZFP57 binding at the wild type *Airn* ICR, and predicted binding in the shuffled *Airn* ICR. **d)** Predicted ZFP57 binding in wild type *Airn* with a scanning 10 bp replacement from the shuffled *Airn* sequence. Each row represents the prediction from one replacement experiments. **e)** ZFP57 binding prediction in shuffled *Airn* + reconstituted ZFP57 binding motifs.

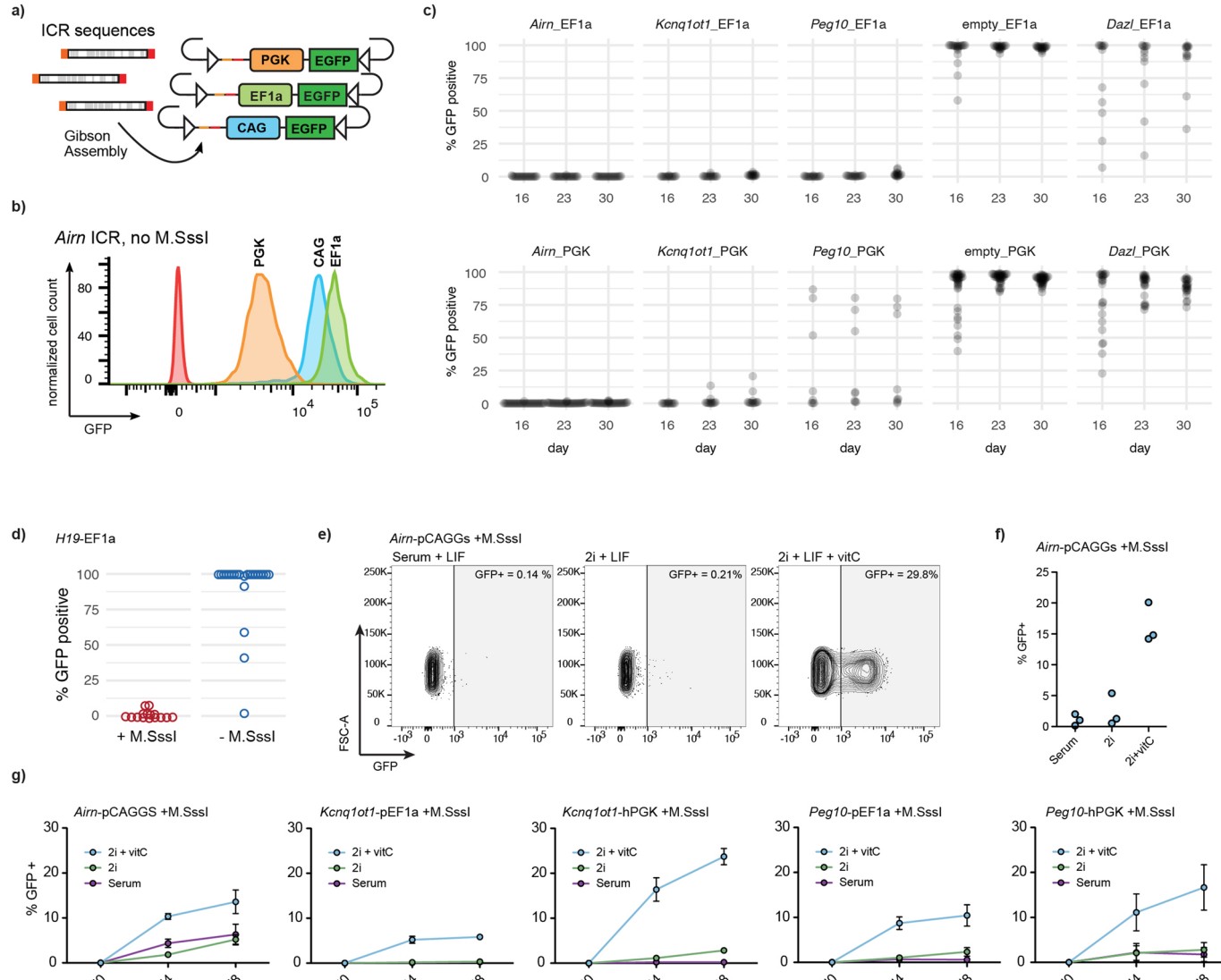

**Extended Data Fig. 5 | Methylated ICR sequences can repress different promoters in *cis*. a**) Design of GFP reporter constructs with different promoters using identical overhang sequences for Gibson assembly, and **b**) flow cytometric analysis of GFP expression of cells that carry the unmethylated, empty reporter constructs without ICRs. **c**) Flow cytometry analysis indicating percentage of GFP-positive cells per population (derived from individual clones) showing stability of repression for methylated ICR reporters in combination with either EF1a (top) or PGK (bottom) promoters measured at 16, 23 and 30 days after transfection. In addition, the GFP percentage is shown for cells receiving reporters with methylated promoters only (no ICRs) or in combination with the *Dazl* promoter as controls. Each dot indicates independently derived clones. **d**) Flow cytometry analysis of cells containing an *H19*-EF1a reporter. **e**) Flow cytometry analysis for a representative mESC clone with the *Airn*-pCAGGS reporter cultured in serum, 2i, or 2i + vitamin C for 12 days. **f**) Flow cytometric analysis of three independent clones with the methylated *Airn*-CAG reporter after 8 days in different media conditions. **g**) Time course for reactivation of different ICR-promoter combinations in different growth conditions. Data points show the mean value of 3 independent clones. Error bar indicates the standard error of the mean.

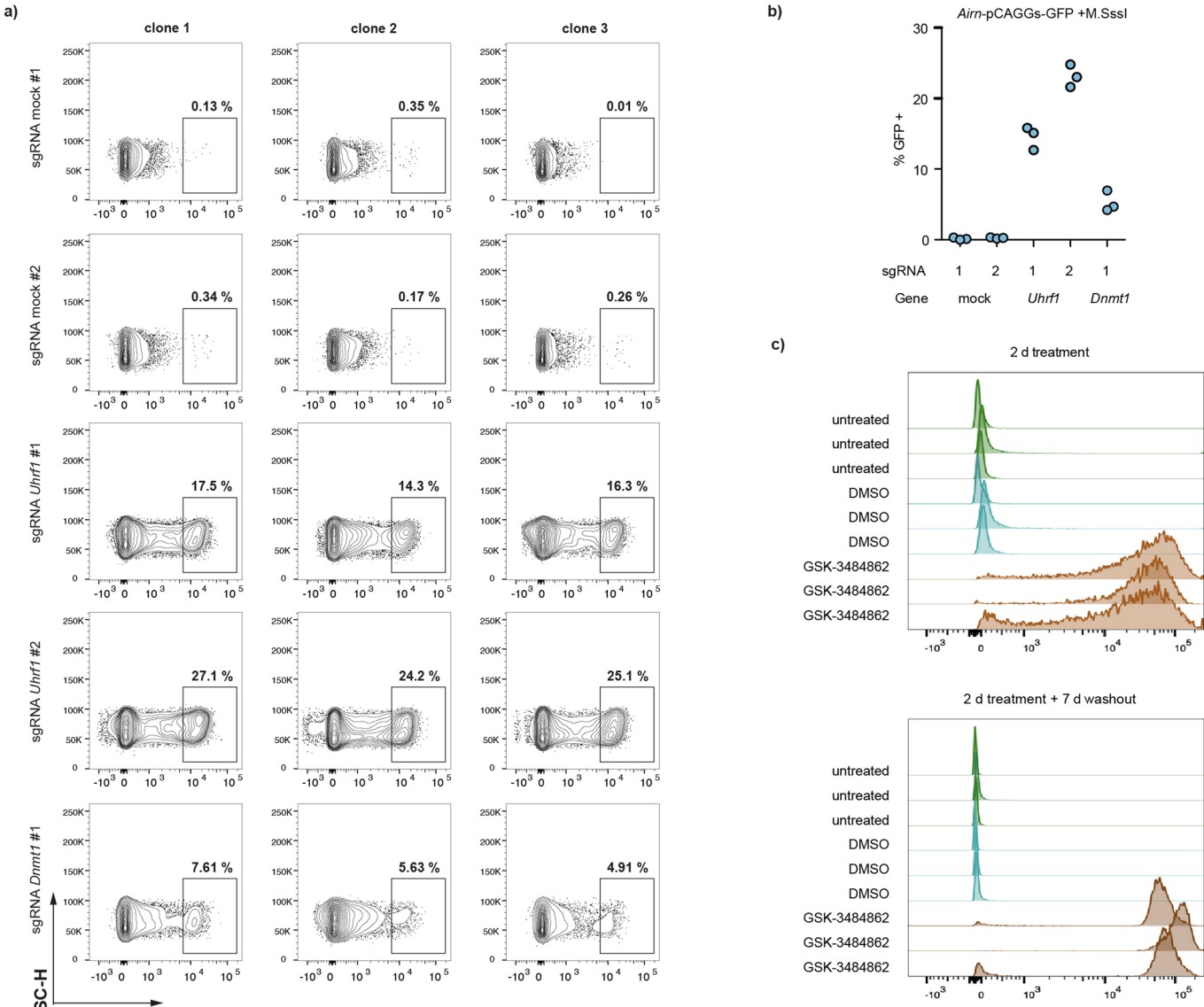

**Extended Data Fig. 6 | ICR sequences repress promoters in a DNA methylation-dependent manner. a)** Flow cytometric analysis of GFP reactivation 8 days after transfection with guide RNAs targeting *Uhrf1* and *Dnmt1* genes, compared to cells transfected with non-targeting guide RNAs. Data is shown from the entire population of targeted cells without pre-selection for KO

cells. **b)** Summarized results from a. Each dot indicates % GFP-positive cells in the entire population of targeted cells. **c)** Flow cytometric analysis of pre-methylated *Airn*-EF1a-GFP treated with the DNA methylation inhibitor GSK-3484862 for 2 days (top) and following washout for 7 days, indicating that once reactivated, the reporter does not re-silence.

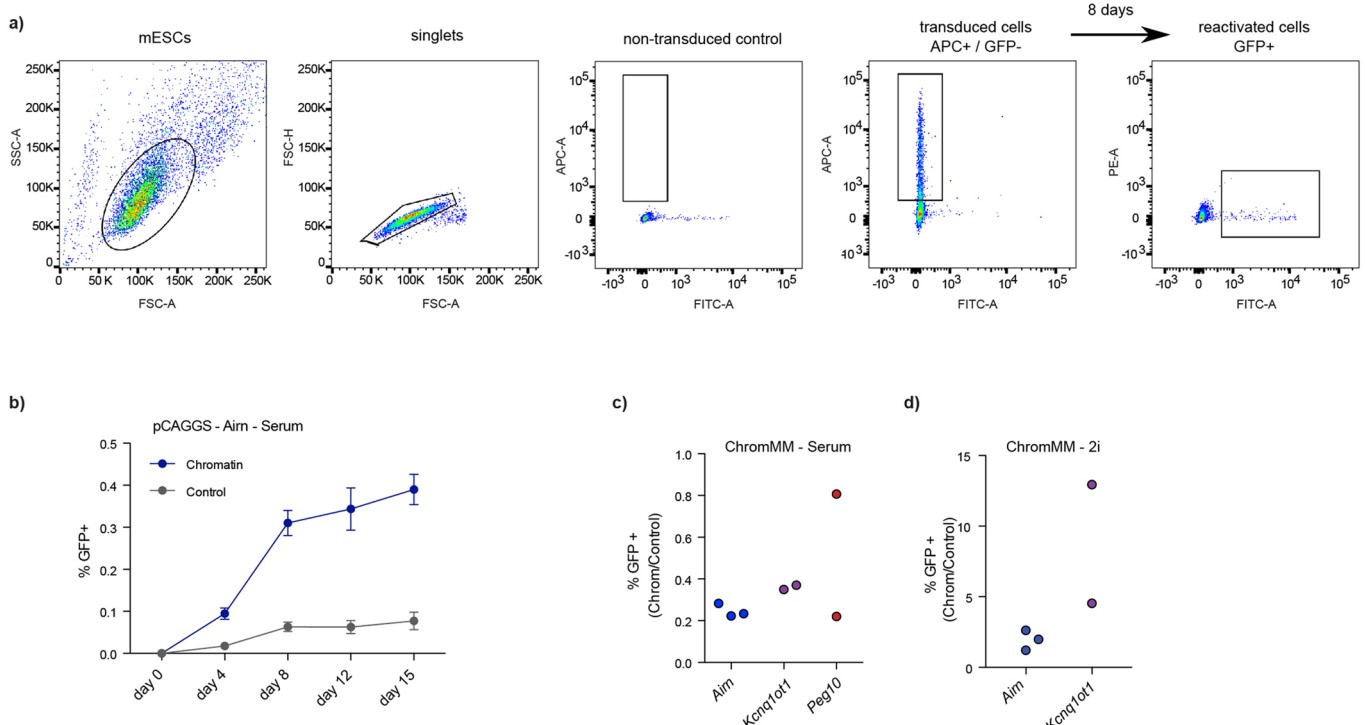

**Extended Data Fig. 7 | Targeted CRISPR screen strategy and setup. a)** FACS gating strategy for CRISPR screens using the ChromMM libraries. Transduced mESCs are selected based on the CD90.1 cell surface marker, co-expressed from the sgRNA containing transgene. Reactivated cells are sorted based on GFP expression (for example 8 days). **b)** Time course experiment for a CRISPR screen using the ChromMM or control library performed with three independent clones. Error bars indicate the standard error of the mean. **c-d)** Flow cytometric analysis of cell lines transduced with the chromatin targeting library vs. the non-targeting control library in serum and 2i, respectively. Axis is indicating percentage of GFP-positive cells in the CRISPR screen population.

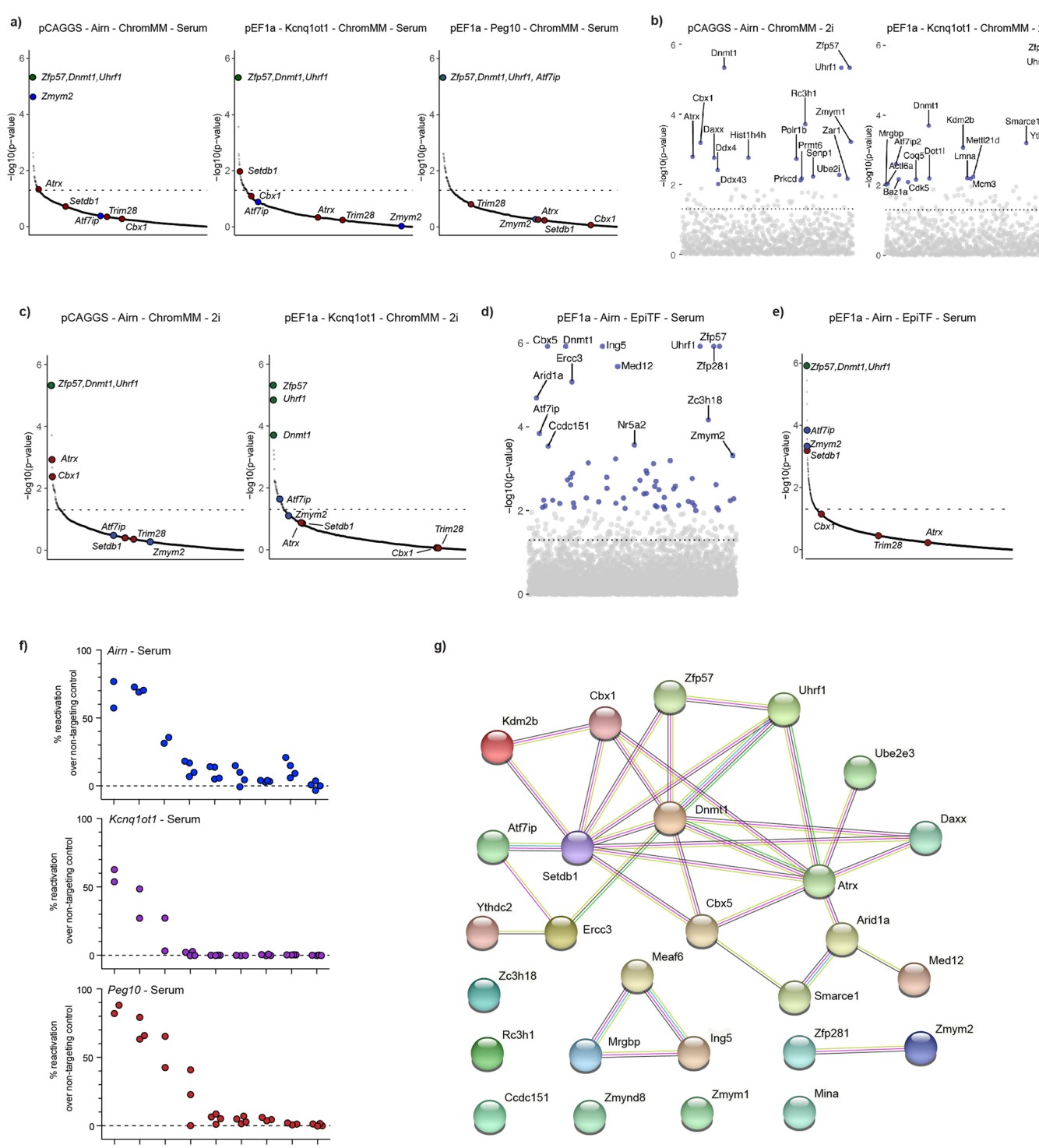

**Extended Data Fig. 8 | See next page for caption.**

**Extended Data Fig. 8 | CRISPR screens identify factors required for methylation maintenance in mESCs. a)** Rank plot for screens in serum conditions. Dashed horizontal line indicates the p-value threshold of 0.05. P-values were obtained using MAGeCK RRA (robust rank aggregation). Red dots indicate known heterochromatin factors associated with ICR regulation, blue dots indicate genes found in multiple screens, and green dots indicate the positive controls. **b)** Overview of CRISPR hits for the *Airn* and *Kcnq1ot1* reporter cell lines grown in 2i conditions. Dashed horizontal line indicates the p-value threshold of 0.05 obtained using MAGeCK RRA (robust rank aggregation). **c)** Rank plot for screens in 2i conditions. Dashed horizontal line indicates the p-value threshold of 0.05 obtained using MAGeCK RRA (robust rank aggregation). **d)** Overview of CRISPR hits using the EpiTF library in the serum-grown *Airn* reporter cell line. Dashed horizontal line indicates the p-value threshold of 0.05 obtained using MAGeCK RRA (robust rank aggregation).

**e)** Rank plot for the screen using the *Airn* pEF1a reporter in serum using the EpiTF library. Dashed horizontal line indicates the p-value threshold of 0.05 obtained using MAGeCK RRA (robust rank aggregation). **f)** Validation of potential candidates using single transfections of guide RNAs against the indicated gene. GFP expression was measured 12 days after transfection. Transfections were performed in technical replicates. Potential candidates were targeted with one independent guide and one guide from the ChromMM library. Data is showing % of positive cells in pools after CRISPR targeting. **g)** Network representation for all potential candidates using the STRING database. Pink edge indicates experimentally determined interactions; cyan edge indicates known interactions from curated databases. Green, red, and blue edges indicate predicted interactions based on gene neighbourhood, gene fusion, and gene co-occurrence, respectively. Yellow and black edges are predicted interactions based on textmining and co-expression, respectively.

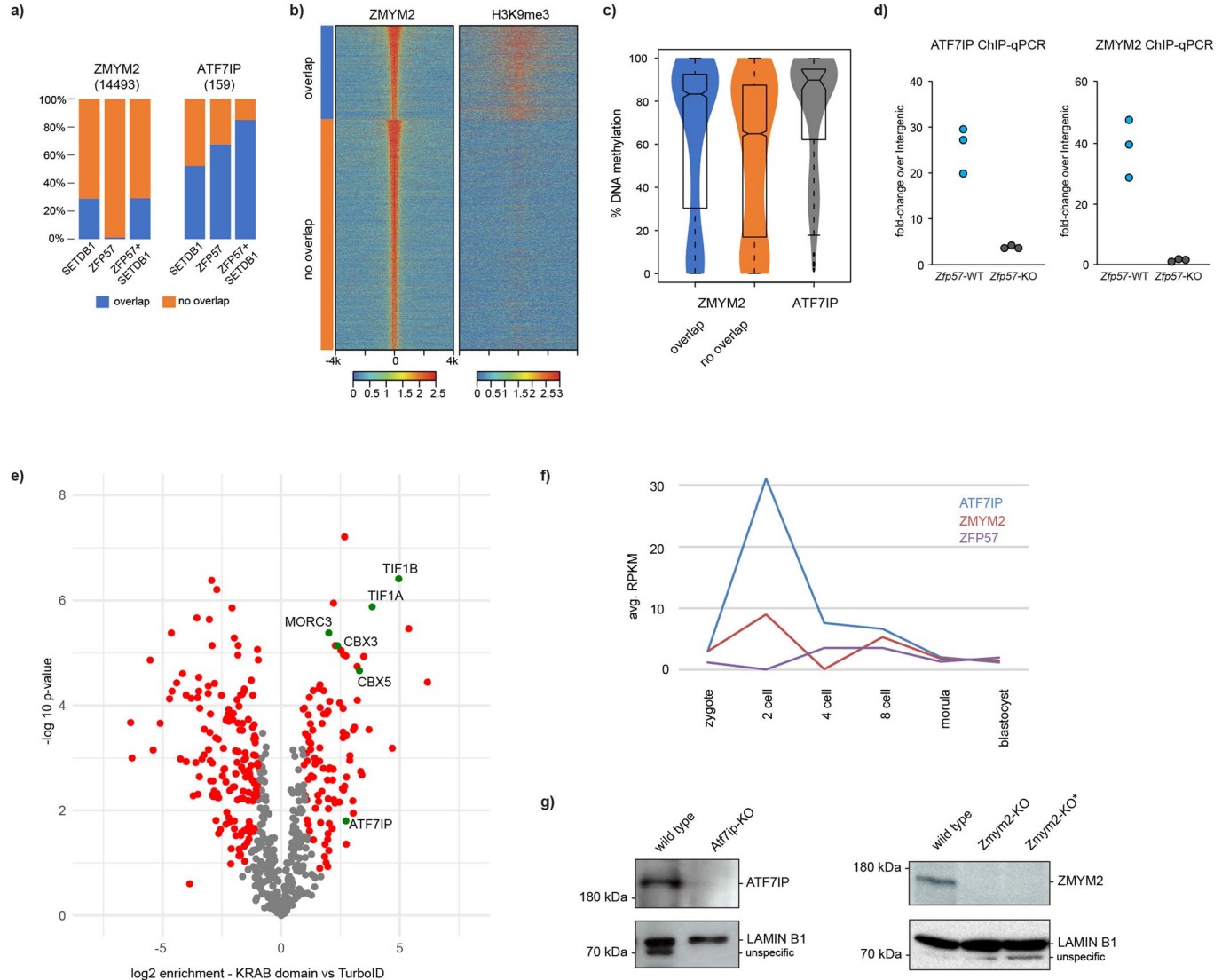

**Extended Data Fig. 9 | ATF7IP and ZMYM2 co-localize with ZFP57 at ICRs.**
**a)** Peak overlap analysis showing percentage of ZMYM2 or ATF7IP peaks
coinciding with SETDB1, ZFP57, or genomic sites co-bound by ZFP57 and SETDB1.
**b)** Heatmap indicating ZMYM2 binding at ZMYM2 peaks and separated by peaks
overlapping SETB1 and SETB1-independent peaks. H3K9me3 ChIP-seq signal
is shown for the same peak sets. **c)** WGBS methylation analysis at independent
ZMYM2 and ATF7IP peaks (N = 159). ZMYM2 peaks are separated by sites
overlapping (N = 4201) and non-overlapping (N = 10292) to SETDB1. Box plots
denote the interquartile range as a box (IQR) and the lowest and highest values
within the range of 1.5 x IQR around the box as whiskers. **d)** ChIP-qPCR for ATF7IP
and ZMYM2 binding at the endogenous *Airn* ICR in wild type and *Zfp57*-KO cells.
ChIP enrichment is normalized to 5% input and calibrated to a background

genomic site (intergenic). Shown are independent technical replicates.
**e)** Volcano plots indicating proteins enriched in the ZFP57_dZNF-TurboID (KRAB
domain only) over a background TurboID cell line (nTurbo). Statistically enriched
proteins are indicated (FDR-corrected two-tailed *t*-test: FDR = 0.05, s0 = 1,
n = 4 technical replicates). **f)** Expression levels for ZFP57, ATF7IP and ZMYM2,
measured at different timepoints during early embryo development. Shown
are RPKM-normalized reads obtained from ref.[73]. **g)** Immunoblot detection of
ATF7IP and ZMYM2 in wild type and KO cells using specific antibodies. Lamin B1
is used as loading control. Asterisk denotes the *Zmym2*-KO clone used for WGBS
analysis. Experiment was repeated at least three times. Molecular weight markers
are indicated.

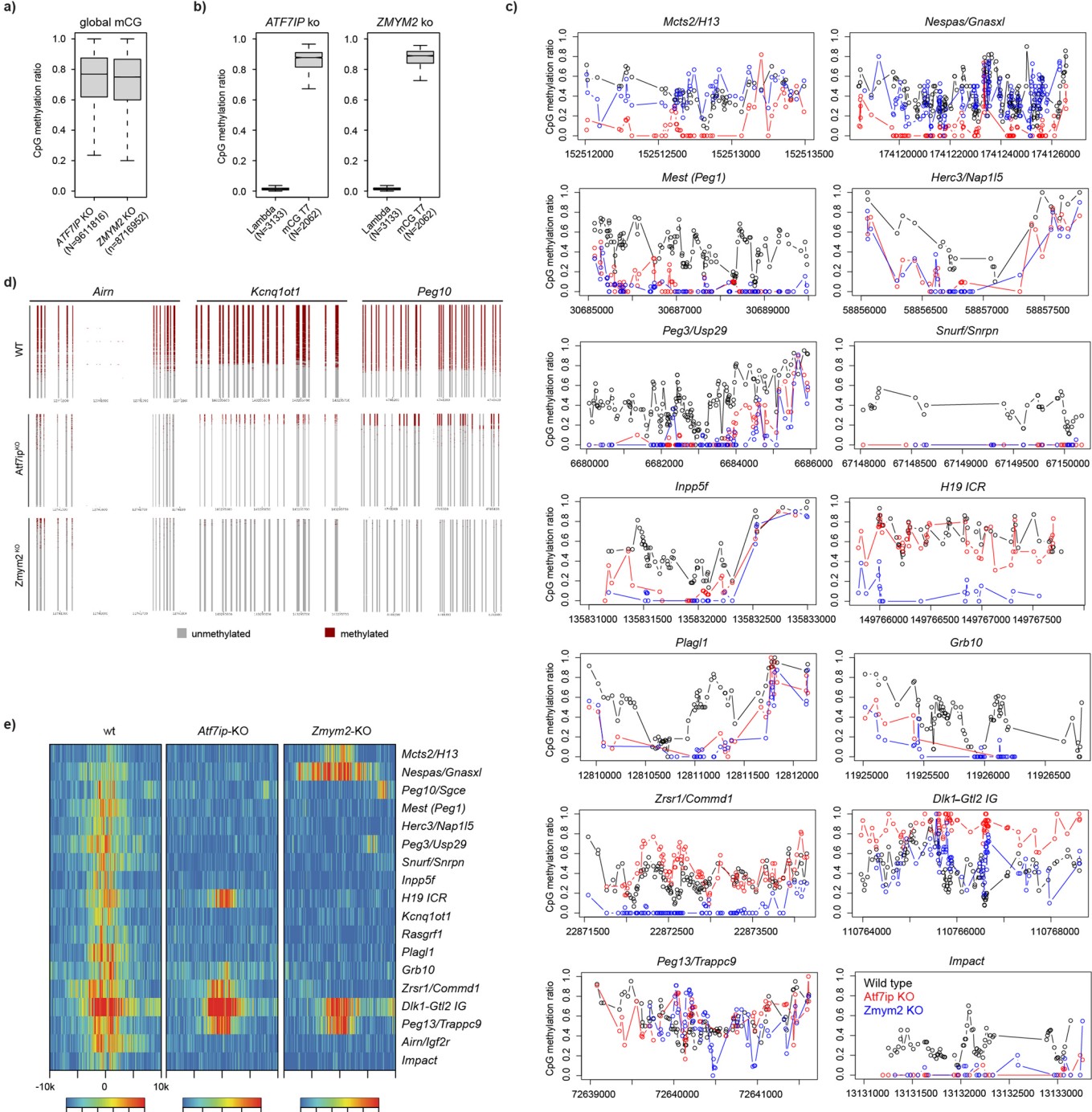

**Extended Data Fig. 10 | DNA methylation and H3K9me3 analysis in Atf7ip and Zmym2 KO cells. a)** Boxplots indicating average DNA methylation over all CpGs covered at least 10x in the mouse genome. Interquartile range is shown as a box (IQR) and the lowest and highest values within the range of 1.5 x IQR around the box as whiskers. Number of independent CpGs analyzed are indicated **b)** Bisulphite conversion controls from spiked-in Lambda and methylated T7 phage DNA show complete conversion of DNA molecules. Interquartile range is shown as a box (IQR) and the lowest and highest values within the range of 1.5 x IQR around the box as whiskers. Number of independent CpGs analyzed are indicated. **c)** WGBS data analysis of all annotated ICRs in wild type, *Atf7ip*-KO, and *Zmym2*-KO mESCs. **d)** Individual methylation profiles from targeted bisulfite sequencing experiments for *Airn*, *Kcnqot1* and *Peg10* ICRs in wild type, *Atf7ip*-KO, and *Zmym2*-KO mESCs. 1000 amplicons per sample were randomly sampled for better visualization. **e)** Heatmap showing H3K9me3 at all ICRs in wild type, *Atf7ip*-KO, and *Zmym2*-KO mESCs. Shown are library-normalized reads per 20 bp.

# Reporting Summary

## Statistics

For all statistical analyses, confirm that the following items are present in the figure legend, table legend, main text, or Methods section.

| n/a | Confirmed | |
|---|---|---|
| ☐ | ☒ | The exact sample size (*n*) for each experimental group/condition, given as a discrete number and unit of measurement |
| ☐ | ☒ | A statement on whether measurements were taken from distinct samples or whether the same sample was measured repeatedly |
| ☐ | ☒ | The statistical test(s) used AND whether they are one- or two-sided *Only common tests should be described solely by name; describe more complex techniques in the Methods section.* |
| ☒ | ☐ | A description of all covariates tested |
| ☐ | ☒ | A description of any assumptions or corrections, such as tests of normality and adjustment for multiple comparisons |
| ☐ | ☒ | A full description of the statistical parameters including central tendency (e.g. means) or other basic estimates (e.g. regression coefficient) AND variation (e.g. standard deviation) or associated estimates of uncertainty (e.g. confidence intervals) |
| ☐ | ☒ | For null hypothesis testing, the test statistic (e.g. *F*, *t*, *r*) with confidence intervals, effect sizes, degrees of freedom and *P* value noted *Give P values as exact values whenever suitable.* |
| ☒ | ☐ | For Bayesian analysis, information on the choice of priors and Markov chain Monte Carlo settings |
| ☒ | ☐ | For hierarchical and complex designs, identification of the appropriate level for tests and full reporting of outcomes |
| ☒ | ☐ | Estimates of effect sizes (e.g. Cohen's *d*, Pearson's *r*), indicating how they were calculated |

*Our web collection on statistics for biologists contains articles on many of the points above.*

## Software and code

Policy information about availability of computer code

| Data collection | For data collection we used standard operating software installed on Illumina MiSeq/NovaSeq sequencers, BD Biosciences FACSCanto II cytometer, BD Bioscience FACSAria III cell sorter, and Thermo Scientific Orbitrap Fusion masspectrometers. |
|---|---|
| Data analysis | For data analysis, the following publicly available tools have been used: FlowJo (10.7) BD FACSDiva (9.1.2) R (4.0.3) BPNet (0.0.22) bowtie 2 (2.3.5.1) trim_galore (0.6.6) picard (2.23.9) MACS2 (2.1.1.20160309) Bismark (0.23.0) Prism (5.0a) QuasR (1.30.0) bedtools (2.27.1) MAGeCK (0.5.9.2) MaxQuant (1.5.3.30) |

For manuscripts utilizing custom algorithms or software that are central to the research but not yet described in published literature, software must be made available to editors and reviewers. We strongly encourage code deposition in a community repository (e.g. GitHub). See the Nature Portfolio guidelines for submitting code & software for further information.

## Data

Policy information about availability of data

All manuscripts must include a data availability statement. This statement should provide the following information, where applicable:

- Accession codes, unique identifiers, or web links for publicly available datasets
- A description of any restrictions on data availability
- For clinical datasets or third party data, please ensure that the statement adheres to our policy

Following published datasets were used in this study:
WGBS - https://www.ncbi.nlm.nih.gov/geo/query/acc.cgi?acc=GSE30206
H3K9me3, H3K4me3, H3K36me3 and H3K27me3 - https://www.ncbi.nlm.nih.gov/geo/query/acc.cgi?acc=GSE23943
DNase-seq and RNA-seq - https://www.ncbi.nlm.nih.gov/geo/query/acc.cgi?acc=GSE67867
SETDB1 - https://www.ncbi.nlm.nih.gov/geo/query/acc.cgi?acc=GSE126243
ZFP57 - https://www.ncbi.nlm.nih.gov/geo/query/acc.cgi?acc=GSE123942
ZMYM2 - https://www.ncbi.nlm.nih.gov/geo/query/acc.cgi?acc=GSE119820
ATF7IP - https://www.ncbi.nlm.nih.gov/bioproject/PRJNA664286

Newly-generated genomics data has been deposited to NCBI GEO under the following accession number GSE176461;
Newly-generated mass spectrometry proteomics data have been deposited to the ProteomeXchange Consortium via the dataset identifier PXD034918.

# Field-specific reporting

Please select the one below that is the best fit for your research. If you are not sure, read the appropriate sections before making your selection.

☒ Life sciences ☐ Behavioural & social sciences ☐ Ecological, evolutionary & environmental sciences

For a reference copy of the document with all sections, see nature.com/documents/nr-reporting-summary-flat.pdf

# Life sciences study design

All studies must disclose on these points even when the disclosure is negative.

| | |
|---|---|
| Sample size | No statistical method was used to pre-determine sample size. |
| Data exclusions | Genomic regions with low-mappability scores (defined by ENCODE) were excluded from further analysis. |
| Replication | All experiments were confirmed by replication at least once with independent cell lines, except for the EpiTF CRISPR screen which itself was intended to be used as an independent replication for the screens performed with the ChromMM library. The exact handling of replicates is depicted in figure panels or the methods section. |
| Randomization | Not relevant for this genomics experiments performed here. Samples were allocated to either wild type or mutant and processed in parallel. Form MS measurements, block-randomization was applied. |
| Blinding | Blinding not relevant for this study. All samples were processed through identical analysis pipelines in parallel. |

# Reporting for specific materials, systems and methods

We require information from authors about some types of materials, experimental systems and methods used in many studies. Here, indicate whether each material, system or method listed is relevant to your study. If you are not sure if a list item applies to your research, read the appropriate section before selecting a response.

## Materials & experimental systems

| n/a | Involved in the study |
|---|---|
| ☐ | ☒ Antibodies |
| ☐ | ☒ Eukaryotic cell lines |
| ☒ | ☐ Palaeontology and archaeology |
| ☒ | ☐ Animals and other organisms |
| ☒ | ☐ Human research participants |
| ☒ | ☐ Clinical data |
| ☒ | ☐ Dual use research of concern |

## Methods

| n/a | Involved in the study |
|---|---|
| ☐ | ☒ ChIP-seq |
| ☐ | ☒ Flow cytometry |
| ☒ | ☐ MRI-based neuroimaging |

# Antibodies

| | |
|---|---|
| Antibodies used | The following antibodies were used for FACS:<br>monoclonal APC-conjugated anti-CD90.1 Invitrogen #17-0900-82 - clone HIS51<br><br>The Following antibodies were used for ChIP:<br>polyclonal anti-H3K9me3 abcam #ab8898<br>polyclonal anti-H3K4me2 Diagenode, C15410035<br>polyclonal anti-ATF7IP Bethyl, A300-169A<br>polyclonal anti-ZMYM2 Bethyl, A301-711A<br><br>The Following antibodies were used for WB:<br>anti-LaminB1 Santa Criuz, sc-3019<br>anti-ATF7IP Bethyl, A300-169A<br>anti-ZMYM2 Bethyl, A301-711A<br>Pierce goat anti-rabbit IgG (H+L) HRP-conjugated (ThermoFischer, 31466),<br>Pierce goat anti-mouse IgG (H+L) HRP-conjugated (ThermoFischer, 31431). |
| Validation | All antibodies are commercial antibodies and validated by the provider and numerous publications.<br><br>FACS<br>monoclonal APC-conjugated anti-CD90.1 Invitrogen #17-0900-82 - clone HIS51:<br>https://www.thermofisher.com/antibody/product/CD90-1-Thy-1-1-Antibody-clone-HIS51-Monoclonal/17-0900-82<br><br>ChIP<br>polyclonal rabbit anti-H3K9me3 (Abcam, ab8898, 5µg/IP): http://www.histoneantibodies.com/FinalArrayData/H3K9me3/<br>polyclonal anti-H3K4me2 Diagenode, C15410035 (5 µg/IP): http://www.histoneantibodies.com/FinalArrayData/H3K4me2/<br>anti-ATF7IP Bethyl, A300-169A:<br>https://www.thermofisher.com/antibody/product/MCAF-Antibody-Polyclonal/A300-169A-M<br>anti-ZMYM2 Bethyl, A301-711A:<br>https://www.fortislife.com/products/primary-antibodies/rabbit-anti-znf198-antibody/BETHYL-A301-711<br><br>WB<br>anti-LaminB1 Santa Criuz, sc-3019:<br>https://www.scbt.com/p/lamin-b1-antibody-c-12<br>anti-ATF7IP Bethyl, A300-169A:<br>https://www.thermofisher.com/antibody/product/MCAF-Antibody-Polyclonal/A300-169A-M<br>anti-ZMYM2 Bethyl, A301-711A:<br>https://www.fortislife.com/products/primary-antibodies/rabbit-anti-znf198-antibody/BETHYL-A301-711 |

# Eukaryotic cell lines

Policy information about cell lines

| | |
|---|---|
| Cell line source(s) | TC-1: Lienert et al., 2011. DOI: 10.1038/nature10716<br>HEK293T: ATCC RRID: CVCL_0063<br>MEL: Feng et al., 2001 DOI:10.1128/MCB.21.1.298-309.2001 |
| Authentication | Genotype confirmed by PCR/Sanger sequencing and Western blot. |
| Mycoplasma contamination | Tested negative |
| Commonly misidentified lines<br>(See ICLAC register) | No commonly misidentified cell lines were used. |

# ChIP-seq

## Data deposition

☒ Confirm that both raw and final processed data have been deposited in a public database such as GEO.

☐ Confirm that you have deposited or provided access to graph files (e.g. BED files) for the called peaks.

| | |
|---|---|
| Data access links<br>*May remain private before publication.* | https://www.ncbi.nlm.nih.gov/geo/query/acc.cgi?acc=GSE176461 |
| Files in database submission | ChIP-seq data for H3K9me3 in ES cells |
| Genome browser session<br>(e.g. UCSC) | no longer applicable |

## Methodology

| | |
|---|---|
| Replicates | one replicate per clone performed |
| Sequencing depth | Over 20 Mio reads aligned once after deduplication and filtering of reads with MAPQ >40 |
| Antibodies | polyclonal anti-H3K9me3 Bethyl, A300-169A (5 µg/IP) |
| Peak calling parameters | No peaks were called on the generated H3K9me3 data |
| Data quality | Sequencing quality was assessed using QuasR qQCReport() |
| Software | trim_galore 0.6.6<br>QuasR 1.30.0 |

# Flow Cytometry

## Plots

Confirm that:

☒ The axis labels state the marker and fluorochrome used (e.g. CD4-FITC).

☒ The axis scales are clearly visible. Include numbers along axes only for bottom left plot of group (a 'group' is an analysis of identical markers).

☒ All plots are contour plots with outliers or pseudocolor plots.

☒ A numerical value for number of cells or percentage (with statistics) is provided.

## Methodology

| | |
|---|---|
| Sample preparation | Cells were harvested by trypsinisation and resuspended in 2 % FCS in DPBS for analysis. If necessary, cells were incubated with monoclonal APC-conjugated anti-CD90.1 Invitrogen #17-0900-82 (1 µl per 15 million cells) for 30 min at 4 ℃. Samples were analysed for eGFP, mScarlet, and APC expression by flow cytometry on a FACSCanto II (BD Biosciences), LSR Fortessa (BD Biosciences), or FACSAria III cell sorter (BD Biosciences). |
| Instrument | FACSCanto II (BD Biosciences), LSR Fortessa (BD Biosciences), or FACSAria III cell sorter (BD Biosciences). |
| Software | Data was aquired with the BD FACSDiva Software (BD Biosciences) and analysed with FlowJo 10.7 |
| Cell population abundance | Purity was assessed for the first sample of a sort. |
| Gating strategy | Gating was performed based on WT cells gated for single cells, using forward scatter area (FSC-A) vs. side scatter area (SSC-A), followed by FSC-A vs. forward scatter height (FSC-H). Gating for eGFP, mScarlet, and APC was done with the help of single-stained or single-fluorescent cells. |

☒ Tick this box to confirm that a figure exemplifying the gating strategy is provided in the Supplementary Information.

