## [Peer Review File · Nature Genetics]

Peer Review Information

Manuscript Title: DNA sequence confers epigenetic bistability at imprinting control regions

Corresponding author name(s): Professor Tuncay Baubec

Reviewer Comments & Decisions:

Decision Letter, initial version:

2nd Aug 2021

Dear Tuncay,

Your Article, "DNA sequence confers epigenetic bistability at imprinting control regions" has now been seen by 3 referees. I sincerely apologize for the long review process.

You will see from the reviewers' comments copied below that while they find your work of considerable potential interest, they have raised quite substantial concerns that must be carefully addressed. Overall, the reviewers' views closely align with our initial editorial assessment, which I previously shared with you. Briefly, the reviewers think that the manuscript reports some interesting findings but that the level of novelty and mechanistic insight is limited at this stage. In our earlier correspondence, you mentioned that there are several ongoing experiments that may address these concerns: further dissection of the ICR DNA elements and introduction of synthetic ICRs into the ES cell genome to identify critical features, neuronal-network-based interpretation of ICR DNA sequence using BPNet along with in silico evaluation of base substitutions and experimental tests using the presented RMCE setup, a new set of ChIP-seq and RNA-seq experiments to address changes upon removal of ATF7IP and ZMYM2, and targeted quantitative mass spectrometry to identify direct regulators at methylated ICRs (using BioID). We are encouraged by these potential developments and strongly encourage you to include these data in a revised manuscript, in addition to new data that fully addresses the reviewers' suggestions/requests.

In sum, in light of the reviewers' comments, we cannot accept the manuscript for publication at this stage, but would be interested in considering a substantially revised version that addresses their serious concerns.

We hope you will find the referees' comments useful as you decide how to proceed. If you wish to submit a substantially revised manuscript, please bear in mind that we will be reluctant to approach the referees again in the absence of major revisions.

If you choose to revise your manuscript taking into account all reviewer and editor comments, please highlight all changes in the manuscript text file. At this stage we will need you to upload a copy of the manuscript in MS Word .docx or similar editable format.

We are committed to providing a fair and constructive peer-review process. Do not hesitate to contact me if there are specific requests from the reviewers that you believe are technically impossible or unlikely to yield a meaningful outcome. If you wish, I'd be happy to discuss the reviewers' comments in detail.

*2) If you have not done so already please begin to revise your manuscript so that it conforms to our Article format instructions, available [here](http://www.nature.com/ng/authors/article_types/index.html). Refer also to any guidelines provided in this letter.

[REDACTED]

If you wish to submit a suitably revised manuscript we would hope to receive it within ~6 months. If you cannot send it within this time, please let us know. We will be happy to consider your revision after 6 months so long as nothing similar has been accepted for publication at Nature Genetics or published elsewhere. Should your manuscript be substantially delayed without notifying us in advance and your article is eventually published, the received date would be that of the revised, not the original, version.

Thank you for the opportunity to review your work.

Sincerely,

Tiago

Tiago Faial, PhD
Senior Editor
Nature Genetics
<https://orcid.org/0000-0003-0864-1200>

Reviewers' Comments:

Reviewer #1:

Remarks to the Author:

Genomic imprinting is regulated by germline derived differentially methylated imprinting control regions (ICRs) that are recognized and maintained in the early embryo and are stable through subsequent development. Understanding the hierarchy of epigenetic events that gives rise to the differential behaviour of the two parental ICRs remains to be fully understood. In this work by Butz and colleagues, underlying ICR properties associated with the maintenance of DNA methylation at imprinting control regions (ICRs) in an ES cell context are elegantly considered. First, a new model is described and dissected to explore the relationship between ICR sequence, ICR methylation and reporter expression using integrated exogenous methylated or unmethylated ICR regions, appropriate controls and a 'shuffled' Airn ICR. The first part of the manuscript describes the faithful behaviour of their model exogenous ICRs, and, with the shuffled locus, the importance of sequence around the CpGs in the maintenance of the methylated state.

The second part of the manuscript uses these exogenous imprints and presents a screen for novel

3factors influencing the faithful epigenetic maintenance of their function. This second part is the section that really has the potential to advance the mechanistic questions that they set out to address. Over the past decades, various ICR-containing transgenes have been inserted within mammalian genomes, usually *in vivo*, and their maintenance assessed. Some findings are consistent with work reported here in terms of faithful behaviour, and others show less consistent results; to my knowledge, none have identified new players in the epigenetic control of ICRs. So the elegant model system described here is welcome. Although proving the faithfulness/usefulness of the model (figures 1-3) is essential in order to understand the implementation of the screen, the findings in the first part are not surprising. It is the section concerning the screen for novel regulators (figures 4-5) that is innovative and has the potential to be impactful and deliver mechanistic insight, particularly into the relationship between DNA methylation and H3K9me3 at imprints. I would have liked to have seen more experimental attention to this section and the findings advanced some more.

Major comments:

1. ICRs can uniquely maintain parental allele-specific methylation. Dnmt1/Uhrf1 and Zfp57/Zfp445/KAP1 are recognized as key factors. Since the shuffled ICR disrupts the ZFP57 binding motifs, it is not surprising therefore that the methylation is lost at this perturbed ICR and hence enthusiasm for the conclusion around the importance of sequence is dampened somewhat. The manuscript would benefit from the inclusion of a shuffled locus which retains the ZFP57 binding sites to conclusively address whether ZFP57 binding is the sole sequence-driven determinant in the maintenance of ICRs.
2. AFTF7IP and ZMYM2 are novel factors identified in the screen and they co-localize at the three ICRs as shown in Figure 5. However, it is not shown whether AFTF7IP and ZMYM2 bind at specific or all ICRs. The authors should indicate (a) which ICRs are bound by AFTF7IP and/or ZMYM2 in ES cells? (b) Alongside this, it is important to also show which regions have co-localization of ZFP57 and/or SETDB1 and which regions are uniquely bound by AFTF7IP or ZMYM2 (independent of ZFP57 and SETDB1). This will provide insight into the hypothesis proposed in the discussion about the function of AFTF7IP and ZMYM2 in the stabilization of the H3K9me3 state and to have a perspective on the hierarchical regulation of DNA methylation/H3K9me3 at imprints (and elsewhere). (c) Furthermore, maybe I missed this but it is unclear from the data if AFTF7IP and ZMYM2 bind in a methylation-sensitive manner – presumably they do. Please can this be assessed (d) It would therefore also be useful to show if regions bound by AFTF7IP or ZMYM2 independently of ZFP57 and Setdb1 are methylated in mES cells.
3. In the ZFP57 KO cells, is the binding of the two novel factors affected at exogenous and endogenous ICRs?
4. It is noteworthy that *in vivo*, Peg10 imprinting is not regulated by ZFP57/ZFP445 – it appears that AFT7IP is more strongly implicated in the Peg10 ICR. Is this the case? Might this be worth commenting on?
5. Although the authors made AFTF7IP KO ES cells and confirmed methylation loss at three ICRs in cells, the function of ZMYM2 has not been validated. There is also no data to show that these two

4factors actually protect methylation at ICRs in vivo. Therefore, the sentence, stating that 'we identify novel factors that prevent switching between methylated and unmethylated states and validate two of these candidates', is overstated since only one is validated and neither are assessed in vivo. I recommend changing this sentence.

6. Lastly, the authors do not extensively address the underlying question that they are trying to address, which is why ICRs maintain or are protected from gaining methylation on one parental allele and not the other, although their sequence is exactly the same. This potentially exciting second part of the manuscript does not quite go far enough to really address the mechanistic role of the two novel factors that they have identified. The hypotheses outlined in the Discussion speculating about the contribution of these factors to the hierarchical relationship between DNA methylation and the role of H3K9me3 in stabilizing and maintaining these states is sensible and if true would represent a significant advance. For publication in Nature Genetics, extending their preliminary findings in that direction is required.

Technical and minor comments:

1. The authors used bisulfite Sanger sequencing and targeted bisulfite sequencing to quantify methylation levels. Bisulfite Sanger sequencing is not a quantitative method due to cloning bias, especially for intermediately methylated regions. It is important to show if the quantitative methylation analysis is reliable. Does methylation show 50% methylation at endogenous ICRs in their hands using this method?
2. In Figure 3b the authors show that under the PCAGGs promoter, Kcnq1ot1 has elevated GFP expression, suggesting that the promoter overcomes the ICRs epigenetic repression. Does this correlate with methylation levels at the ICR and/or promoter? It would be good to understand the behaviour of this construct.
3. In Figure 3e the extent of re-activation after knockout of Uhrf1 and Dnmt1 is quite low. Gfp is only at 5% in Dnmt1 KOs after 8 days - what does it look like after longer? And how successful are the knockouts on Uhrf1 and Dnmt1 levels? Again, what does the methylation at these promoters now look like?
4. Figure 4b&c- the y-axes say "GFP%" but in 4b the axis goes to 1 and in 4c the axis goes to 15... are they both %? Or is the maximum "1" value in graph 4b equivalent to 100%? The text says that in 2i media (graph 4c) there are more GFP positive cells compared to growth in serum (graph 4b). However this is not clear in the graphs as presented perhaps because I don't understand the axes - for instance for Airn, it looks like there is 0.2 (20%?) expression of GFP in serum and somewhere around 2-3 in 2i. Clarification is required here or perhaps this can be illustrated in a different way.
5. Line 239 should be 'extent' not 'extend'.
6. In Figure 2 the authors note "CpG positions marked with x correspond to unaligned nucleotides". I am not sure what this means. Does the sequence quality fail at that point? Please explain this better.

Reviewer #2:

Remarks to the Author:

In this work, Butz et al. investigate the maintenance of epigenetic states at imprinting control regions

5(ICRs). Integrating methylated and unmethylated sequences of three ICRs into a 'safe-harbor' locus in mouse ESCs, confirmed that ICRs can autonomously maintain their pre-established DNA methylation status, unlike other elements with similar GC content. The authors also observed that the state of DNA methylation is no longer preserved when shuffling the ICR sequence (even though CpGs remain intact), consistent with the notion that maintenance is dependent upon the binding of trans-acting factors (such as Zfp57). The authors then used the autonomous ICRs as reporters and performed a CRISPR/Cas9 screen to identify factors that are involved in DNA methylation maintenance. In addition to known factors like Dnmt1, Uhrf1, and Zfp57, they proposed additional hits and validated the involvement of two of them - Atf7ip and Zmym2 in preserving DNA methylation and H3K9me3 at the three examined ICRs.

Overall, the manuscript comprises two main parts: (i) studying the sequence composition (cis) of ICRs and (ii) identifying trans-acting factors that potentially regulate its inherent epigenetic bistability. While these are clearly fundamental questions, we find that, in its current state, the manuscript falls short in trying to address both topics.

Main concerns:

1. The manuscript in its current form provides only a few new insights into cis determinants that asymmetrically regulate epigenetic states in ICRs. Presumably, a similar approach to Lienert et al. 2011 can be used here to uncover or refine binding motifs (such as for Zfp57), structural organization, and additional critical elements that can preserve the repressed state (even if demonstrated using a single ICR). Another outstanding issue is the minimal distance between the ICR and downstream promoter that is required to maintain repression. This can be addressed by integrating spacers with different GC content between the two elements.

Finally, all tested ICRs are maternally imprinted and hence associated with specific gene promoters. This raises the question of whether paternally imprinted intergenic ICRs can similarly repress an adjacent promoter element?

2. We appreciate that identifying 2 additional candidate genes that regulate the maintenance of ICR methylation is an important contribution to the field. However, current data does not go beyond screen validation. For example, KO of Atf7ip and Zmym2 in mESCs could imply that both of these genes directly bind Zfp57 or are involved in the recruitment of Kap1. This can be addressed using co-IP or other methods. Alternatively, any insights into the expression of Atf7ip and Zmym2 following implantation, in zygotes, and in germ cells, or the effects of KO on imprinting in vivo would significantly increase the impact of this report.

Minor issues:

3. Figure 3. The authors should provide methylation data on the promoter regions integrated downstream of the ICRs to show that it indeed correlates with the methylation status of the ICR.

4. One control that is missing is the integration of a methylated version of the promoter region - without the upstream ICR. This will provide information about the kinetics of demethylation when the repressive element is missing.

65. Can the authors include data validating KO of Atf7ip and Zmym2 at the protein level? If this is not possible, validation at the genomic level should be provided

6. Figure 3e. We find it surprising that KO of either Uhrf1 or Dnmt1 is not sufficient to de-repress the majority of reporter cells after 8 days in culture. Is that because the KO was done on the bulk population and not on validated KO clones? Can the authors explain this?

Reviewer #3:

Remarks to the Author:

Imprint control regions (ICRs) are classic examples of epigenetic regulation, as genetically identical sequence exhibit differential DNA methylation states depending on the parent of origin. Moreover, these “bistable” loci (as the authors put it), can persist in somatic tissues throughout life, thus exhibiting remarkable epigenetic memory. Mouse ESCs are a pertinent model to study ICR regulation: recent studies have shown that epigenetic memory is less stable in ESCs than in differentiated cell types (e.g., <https://www.biorxiv.org/content/10.1101/2021.05.11.443595v1>); however ICRs are an exception, as the factors that protect the regions from losing DNA methylation during epigenetic reprogramming are likely present. In this study, Butz et al. take advantage of this fact to determine the properties of DNA Methylation-based ICR control, and perform a CRISPR-based screen to discover important factors in this process.

In fact, the manuscript is essentially two different studies. In the first part, the authors utilized the recombinase-mediated cassette exchange (RMCE) system to efficiently integrate a set of ICR sequences into the “neutral” β globin locus. This is a strategy that has been used with high success in previous studies from the Schuebeler lab, where the senior author from this study did his postdoctoral work. I found the experiments to be well-executed and convincing: pre-methylated ICRs tend to stay methylated, while unmethylated versions of the same sequences remain DNA methylation free—although it should be noted that in some cases de novo DNA methylation occurs at the unmethylated sequences, and this is stably maintained (Supp. Fig. 1d-f); this is a minor point, but is not commented on in the text. The DNA methylation maintenance does not occur on sequences with similar length, GC content, and CpG density. Moreover, scrambling the sequence between CpGs abolishes DNA methylation maintenance at ICRs. The authors state that this shows that the DNA sequence—in addition to the DNA methylation—that confers its epigenetic properties. These experiments are a nice way to demonstrate this fact, but it is not a novel finding. It has been known for some time that ZFP57 (and more recently ZFP445) recognize both genetic and epigenetic information. Therefore, this is more of a confirmation than a discovery. And as mentioned, given ESCs exhibit epigenetic plasticity, it is again rather expected CpG-rich regions absent these key genetic motifs would not maintain DNA methylation. My presumption is if this same experiment was performed in a differentiated cell-type, the methylated sequences very likely might maintain the mark.

The second part of the paper was a CRISPR based genetic screen taking advantage of the ICR epigenetic stability. By placing pre-methylated ICRs upstream of a strong promoter and a reporter, in some cases the reporter would be silenced. Therefore, mutagenesis screens with various ICR/reporter

7combinations could identify key regulators of ICR maintenance. Indeed, DNA methylation maintenance factors Dnmt1 and Uhrf1 were enriched in the screens, as well as Zfp57. The authors focused on two more factors: ATF7IP and ZMYM2. Beyond a ChIP-seq experiment, there was very little characterization of the factors, however. The superficiality of this portion of the paper made it feel somewhat shoe-horned in.

In summary, while the first part of the paper is strong, it suffers from lack of novelty. The second part could potentially lead to interesting directions with more in depth experiments, but those were not performed. Below I have added some minor comments that could strengthen the paper for a near-term submission:

Comments

- If the ESCs are differentiated, and the RMCE experiment is re-performed, is the genetic component of the ICRs less important? For example, would the shuffled Airn ICR maintain its methylation better out of an ESC context. I think this would be important to show how important genetic differences would be for ESCs/epigenetic reprogramming, but the epigenetic signature is more important at later stages of differentiation/development.
- 1b - Tci is not apparently protected from de novo methylation (45.9% methylated in SssI -), and then the methylated sequence gets eroded. What is the nature of this erosion? Do molecules look methylated and unmethylated, or it exhibits a "salt and pepper". This jumped out to me because it seems to fit an intermediate category of DNA methylation not discussed.
- Supp. Fig. 1d-f: The de novo + maintenance property of some of the unmethylated ICRs should be discussed. Is this just a random event that occurs upon integration at the RMCE site?
- What promoters were finally used for the three ICRs in the screen? This is minor, but it is not clear in the manuscript
- The characterization of ZMYM2 (in particular) and ATF7IP could have gone much deeper. Here are some quick ideas:
 - What is the global methylation state in the mutants? This could be done by a LUMA assay, or potentially a whole-genome sequencing approach (e.g., RRBS or WGBS)
 - What is the impact of methylation maintenance of ICRs in a Zmym2 mutant? If these cells grow slowly, this could be done with a degron system for example.
 - The ChIP seq data should be analyzed more thoroughly. For starters, the binding status at all ICRs should be discussed.

Author Rebuttal to Initial comments

Point by point response to reviewer's comments:

We thank the reviewers for the constructive criticism they raised during the first evaluation of our manuscript. The major requests were to provide additional analysis of the newly identified factors and show their involvement in regulating ICRs genome-wide. Furthermore, additional requests were to address the behavior of our ICR reporters, and test the contribution of ZFP57 to the observed DNA sequence dependence in ES cells.

We have now addressed all requests in the following point-by-point response and provide a series of new experiments and data analysis to support our responses and conclusions. Notably, we in this new version we have generated 26 new reporter lines in ES and MEL cells by RMCE to further dissect the ICR DNA elements and identify critical features; We utilized neuronal-network-based interpretation of ZFP57 binding in the Airn ICR sequence using BpNet, followed by experimental evaluation of synthetic sequences using the RMCE setup. We generated genome-wide bisulphite sequencing and ChIP-seq datasets in Atf7ip-KO and Zmym2 KO ES cells to evaluate their contribution to the epigenetic memory at all ICRs. And finally, we identify the proteome composition at ZFP57-bound sites using BioID as an orthogonal approach to detect factors enriched at ICRs

The new additions and responses to the reviewer's comments can be found in the point-by-point response below.

Reviewers' Comments:

Reviewer #1:

Remarks to the Author:

Genomic imprinting is regulated by germline derived differentially methylated imprinting control regions (ICRs) that are recognized and maintained in the early embryo and are stable through subsequent development. Understanding the hierarchy of epigenetic events that gives rise to the differential behaviour of the two parental ICRs remains to be fully understood. In this work by Butz and colleagues, underlying ICR properties associated with the maintenance of DNA methylation at imprinting control regions (ICRs) in an ES cell context are elegantly considered. First, a new model is described and dissected to explore the relationship between ICR sequence, ICR methylation and reporter expression using integrated exogenous methylated or unmethylated ICR regions, appropriate controls and a 'shuffled' Airn ICR. The first part of the manuscript describes the faithful behaviour of their model exogenous ICRs, and, with the shuffled locus, the importance of sequence around the CpGs in the maintenance of the methylated state.

The second part of the manuscript uses these exogenous imprints and presents a screen for novel factors influencing the faithful epigenetic maintenance of their function. This second part is the section that really has the potential to advance the mechanistic questions that they set out to address. Over the past decades, various ICR-containing transgenes have been inserted within mammalian genomes, usually in vivo, and their maintenance assessed. Some findings are consistent with work reported here in terms of faithful behaviour, and others show less consistent results; to my knowledge, none have identified new players in the epigenetic control of ICRs. So the elegant model system described here is welcome. Although proving the faithfulness/usefulness of the model (figures 1-3) is essential in order to understand the implementation of the screen, the findings in the first part are not surprising. It is the section concerning the screen for novel regulators (figures 4-5) that is innovative and has the potential to be impactful and deliver mechanistic insight, particularly into the relationship between DNA methylation and H3K9me3 at imprints. I would have liked to have seen more experimental attention to this section and the findings advanced some more.

We thank the referee for their critical comments and highlighting potential impact our study could have on advancing mechanistic insights into the relationship between DNA methylation and H3K9me3 at ICRs.

Major comments:

1. ICRs can uniquely maintain parental allele-specific methylation. Dnmt1/Uhrf1 and Zfp57/Zfp445/KAP1 are recognized as key factors. Since the shuffled ICR disrupts the ZFP57 binding motifs, it is not surprising therefore that the methylation is lost at this perturbed ICR and hence enthusiasm for the conclusion around the importance of sequence is dampened somewhat. The manuscript would benefit from the inclusion of a shuffled locus which retains the ZFP57 binding sites to conclusively address whether ZFP57 binding is the sole sequence-driven determinant in the maintenance of ICRs.

Response #1: We agree with the referee that sequence shuffling of the Airn ICR does not allow to discern effects imposed by loss of ZFP57 binding from other DNA sequence properties. Furthermore, this approach does not address if the ZFP57 motifs are only required or are even sufficient to maintain DNA methylation at the Airn ICR.

We have now addressed these questions by reconstituting the four ZFP57 motifs in the shuffled Airn ICR at their original position and integrated this sequence pre-methylated and unmethylated to the RMCE site in ES cells. In the case of the unmethylated shuffled Airn ICR + ZFP57 motifs, the synthetic ICR fails to retain its hypomethylated state, resulting in random and intermediate methylation. In contrast, the pre-methylated shuffled Airn ICR + ZFP57 motifs faithfully retains its hypermethylated state (see new Figure 2e).

Furthermore, we wanted to test how this shuffled Airn + Zfp57 motifs would influence transcriptional activity from a downstream GFP reporter, therefore we included this DNA sequence in our reporter construct and integrated them as unmethylated and methylated DNA to the RMCE locus. The unmethylated version resulted in stochastic loss of transcriptional activity, while the methylated version led to stable repression of the GFP reporter for prolonged cultivation (New Figure 3c)

We thank the reviewer for suggesting this experiment, which shows that the ZFP57 binding elements are not only required, but also sufficient for the maintenance of DNA methylation and repressive activity of the Airn ICR in mouse ES cells.

2. AFTF7IP and ZMYM2 are novel factors identified in the screen and they co-localize at the three ICRs as shown in Figure 5. However, it is not shown whether AFTF7IP and ZMYM2 bind at specific or all ICRs. The authors should indicate (a) which ICRs are bound by AFTF7IP and/or ZMYM2 in ES cells? (b) Alongside this, it is important to also show which regions have co-localization of ZFP57 and/or SETDB1 and which regions are uniquely bound by AFTF7IP or ZMYM2 (independent of ZFP57 and SETDB1). This will provide insight into the hypothesis proposed in the discussion about the function of AFTF7IP and ZMYM2 in the stabilization of the H3K9me3 state and to have a perspective on the hierarchical regulation of DNA methylation/H3K9me3 at imprints (and elsewhere). (c) Furthermore, maybe I missed this but it is unclear from the data if AFTF7IP and ZMYM2 bind in a methylation-sensitive manner – presumably they do. Please can this be assessed (d) It would therefore also be useful to show if regions bound by AFTF7IP or ZMYM2 independently of ZFP57 and Setdb1 are methylated in mES cells.

Response #2: Related to (a), we have now re-analyzed published ATF7IP and ZMYM2 ChIP-seq datasets and compared their localization with SETDB1 and ZFP57 at all annotated ICRs in the mouse ES cell genome. We observe that all factors co-localize to all annotated ICRs, with minor exceptions for H19 ICRs where we observe reduced ZMYM2 localization in comparison to ATF7IP, and lack of binding of ATF7IP to MCTS2/H13. Besides these ICRs, the general trend is ATF7IP and ZMYM2 colocalization in presence of SETDB1, even at ICRs that show reduced binding of ZFP57 (Plagl1, H13, Nap115). These new results are shown in new Figure 5b.

As suggested in (b), we furthermore expanded this analysis to compare genome-wide binding preferences of the same factors to their binding properties at ICRs. Towards this we called peaks for all four factors (SETDB1, ZFP57, ATF7IP, ZMYM2) and H3K9me3 and investigated overlapping and individual binding. We observe a strong co-localization of ATF7IP with SETDB1 or ZFP57, which is even further increased at sites that are co-bound by ZFP57 and SETDB1 (New Figure 5c). In case of ZMYM2, we observe less colocalization with SETDB1 and very little localization to ZFP57 sites, indicating that ZMYM2 binds to a diverse set of regions in the genome (New Figure 5c). As expected, sites that are co-bound by ZMYM2 and SETDB1 are also enriched for H3K9me3 and DNA methylation compared to sites bound by ZMYM2 alone (new Sup. Fig. 13 a-b).

We also agree that it's important to investigate the DNA-methylation-dependent localization of ATF7IP and ZMYM2, as suggested in (c and d). Based on analysis of DNA methylation under ATF7IP and ZMYM2 peaks we observed increased DNA methylation for ATF7IP sites (which predominantly binds to chromatin together with SETDB1/ZFP57). In case of ZMYM2 we observe binding to methylated DNA at regions co-bound by SETDB1/ZFP57, while regions bound by ZMYM2 only tend to have less DNA methylation (New Sup. Fig. 13b).

3. In the ZFP57 KO cells, is the binding of the two novel factors affected at exogenous and endogenous ICRs?

Response #3: We thank the reviewer for this suggestion. We performed ChIP-qPCR at the endogenous Airn ICR in Zfp57-KO and observe loss of binding of both ATF7IP and ZMYM2 in absence of Zfp57 (New Sup. Fig. 13c). We attempted several times to perform ChIP-seq for ATF7IP and ZMYM2, however the obtained amount of IP material did not allow to generate sequencing libraries.

4. It is noteworthy that in vivo, Peg10 imprinting is not regulated by ZFP57/ZFP445 – it appears that ATF7IP is more strongly implicated in the Peg10 ICR. Is this the case? Might this be worth commenting on?

Response #4: This seems to be indeed the case. We observe ATF7IP to be highly ranked when using the Peg10 reporter in our screens, with similar ranking scores as observed for DNMT1 or UHRF1 – which was not observed for the other ICR reporters. This goes in line with a recent report that suggests ATF7IP to be a specific regulator of the Peg10 imprint in human ES cells (PMID: 34795250). However, and in contrast to human ESCs, murine ATF7IP influences DNA methylation at all tested endogenous ICRs, except H19, Meg3 and Peg13, suggesting that ATF7IP is not specific to Peg10 only.

Furthermore, we observe ZFP57 to be highly enriched in the Peg10-ICR-reporter screen (Figure 4b) and we also identify ATF7IP to colocalize to ZFP57-bound regions by ChIP-seq (Figure 5b) and further by mass-spectrometry (Figure 5d). This discrepancy in ZFP57 contribution to Peg10 may reflect the genomic context of the Peg10 ICR in vivo vs. our reporter setup that uses only the “solo” ICR at a remote location. A potential explanation is that loss of ZFP57 may be buffered by other regulatory mechanisms at the endogenous Peg10 site. However, when the ICR is isolated from that context, we can identify that its core regulatory circuitry is indeed dependent of ZFP57.

5. Although the authors made ATF7IP KO ES cells and confirmed methylation loss at three ICRs in cells, the function of ZMYM2 has not been validated. There is also no data to show that these two factors actually protect methylation at ICRs in vivo. Therefore, the sentence, stating that ‘we identify novel factors that prevent switching between methylated and unmethylated states and validate two of these candidates’, is overstated since only one is validated and neither are assessed in vivo. I recommend changing this sentence.

Response #5: We have now generated Zmym2 KO cells and have included whole genome bisulphite datasets for Atf7ip-KO and Zmym2-KO ESCs. These enabled us to analyze the DNA methylation status of all ICRs in the mouse genome. In both KO cell lines DNA methylation is drastically reduced at the majority of ICRs indicating that these factors play a role in protection from loss of methylation at ICRs. The only exceptions are Meg3/Rian and Peg13, which retain methylation in both KO cells. In addition, we see that Gnas/Nespas and H13/Mct5 lost DNA methylation only in absence of ATF7IP; while Zrsr1/Commd1 and H19 lost DNA methylation in absence of Zmym2. This new data is shown in the new figures 6a and Sup. Fig. 14c.

In addition, we have also included independent measurements using targeted bisulphite PCR for Peg10, Airn and Kcnqot1 ICRs in both mutant cell lines to confirm the results obtained from WGBS. This new data is shown in Sup. Figure 14d.

We agree that we are not showing mechanisms used by these factors to actively protect from loss of methylation in vivo, but rather that their absence leads to a decrease in DNA methylation (and H3K9me3) at ICRs. We have now changed our statement accordingly to also avoid confusion: “we identify novel factors that prevent switching from methylated to unmethylated states and show that two of these candidates, ATF7IP and ZMYM2, are important for epigenetic memory at ICRs in embryonic stem cells”.

6. Lastly, the authors do not extensively address the underlying question that they are trying to address, which is why ICRs maintain or are protected from gaining methylation on one parental allele and not the other, although their sequence is exactly the same. This potentially exciting second part of the manuscript does not quite go far enough to really address the mechanistic role of the two novel factors that they have identified. The hypotheses outlined in the Discussion speculating about the contribution of these factors to the hierarchical relationship between DNA methylation and the role of H3K9me3 in stabilizing and maintaining these states is sensible and if true would represent a significant advance. For publication in Nature Genetics, extending their preliminary findings in that direction is required.

Response #6:

We thank the reviewer for pointing this out. In this version we provide a large set of new data and evidence to support our initial hypothesis. First, we show that ATF7IP and ZMYM2 are colocalizing to ICRs together with ZFP57 and SETDB1. The colocalization of ZFP57 and ATF7IP on chromatin we further confirm through biochemical assays using proximity biotinylation and detection mass spectrometry. Furthermore, we show the global and ICR-specific effects of ZMYM2 or ATF7IP depletion in ES cells. These result in reduced DNA methylation and H3K9me3 at the majority of ICRs, without having a strong global effect of these marks outside of ICRs. Altogether, these results support our previous hypothesis and are now further described in the new figures

5 and 6 and associated Supplemental Figures 13-15 of the revised manuscript. We hope the reviewer agrees that these novel additions sufficiently support our initial hypothesis.

Technical and minor comments:

1. The authors used bisulfite Sanger sequencing and targeted bisulfite sequencing to quantify methylation levels. Bisulfite Sanger sequencing is not a quantitative method due to cloning bias, especially for intermediately methylated regions. It is important to show if the quantitative methylation analysis is reliable. Does methylation show 50% methylation at endogenous ICRs in their hands using this method?

Response #7: All bisulphite PCR results shown in the figures are using unique sequences analyzed using QUMA (PMID: 18487274). We have also measured bisulphite PCR amplicons by high-throughput sequencing and we observe that methylated vs. unmethylated ICR molecules are represented in a range between 40 and 60%, indicating that this method is indeed representing the expected methylation (Supplementary Figure 14 d). Furthermore, in this work we predominantly focus on the ICRs that are integrated at the RMCE site and therefore we don't require to discriminate between two differentially-methylated alleles.

2. In Figure 3b the authors show that under the PCAGGs promoter, Kcnq1ot1 has elevated GFP expression, suggesting that the promoter overcomes the ICRs epigenetic repression. Does this correlate with methylation levels at the ICR and/or promoter? It would be good to understand the behaviour of this construct.

Response #8: Indeed, the strong CAGGS promoter seems to overcome DNA methylation at the Kcnq1ot1 and Peg10 ICRs. It is worth noting that the synthetic CAGGS promoter is a very strong CpG island promoter and therefore we excluded Peg10 and Kcnq1ot1 constructs containing CAGGS promoters from further analysis and screens.

However, we have included the following analysis for the reviewer in this response: To have a closer look at the behavior of the constitutive promoters in combination with the methylated ICRs, we have now performed a time-course of populations derived from newly generated clones and measured the percentage of GFP-positive cells over time. In our study this was mainly aimed for PGK and EF1A (shown in new Sup. Fig. 8c), but we include here the results for the CAGGS-Kcnq1ot1 and Peg10 ICRs (Rebuttal Figure 1).

This analysis allowed us to test if the observed loss of silencing, as measured by GFP-positive cells, is at a steady-state or increasing/decreasing. Unlike PGK and EF1A promoters that show a stable repression at all timepoints (Sup. Fig. 8c), CAGGS-Kcnq1ot1 and CAGGS-Peg10 show a gradual loss of silencing in all clonal populations, resulting in full GFP reactivation after 30 days.

Rebuttal Figure 1: Flow cytometric analysis indicates GFP activity in independent cell lines retrieving methylated RMCE donor plasmids containing reporters using the CAGGS promoter alone, or in combination with ICRs. GFP activity was measured at three consecutive time points (16, 23 and 30 days).

While important for the initial characterization for all reporters, we find that since we do not follow up on the CAGGS-Kcnq1ot1 or -Peg10 ICR reporters, any further dissection of this construct would distract too much from the main results.

3. In Figure 3e the extent of re-activation after knockout of Uhrf1 and Dnmt1 is quite low. Gfp is only at 5% in

Dnmt1 KOs after 8 days - what does it look like after longer? And how successful are the knockouts on *Uhrf1* and *Dnmt1* levels? Again, what does the methylation at these promoters now look like?

Response #9: The representation is showing % GFP-positive cells from a bulk population targeted with CRISPR-Cas9 against the indicated gene of interest. This results in a mix of cells harboring homozygous, heterozygous mutations, or homozygous wild type alleles which are analyzed by FACS. Due to varying levels of KO success, this composition can vary from attempt to attempt and leads to the reduced number of GFP-positive cells in the population. However, the FACS histograms, which we include now as Sup. Figure 9a, indicate full reactivation of the GFP reporter in cells where the KO was successful.

Since this approach leads to variable results depending on KO efficiency, we have now used the novel DNA methylation inhibitor GSK-3484862 to test the contribution of DNA methylation to reporter-repression. We observe that already after 2 days, the reporters become activated, with over 90% GFP-positive cells, indicating that the reporter responds to loss of DNA methylation. These new results are shown in Figure 3d and Sup. Figure 9c.

*4. Figure 4b&c- the y-axes say "GFP%" but in 4b the axis goes to 1 and in 4c the axis goes to 15... are they both %? Or is the maximum "1" value in graph 4b equivalent to 100%? The text says that in 2i media (graph 4c) there are more GFP positive cells compared to growth in serum (graph 4b). However this is not clear in the graphs as presented perhaps because I don't understand the axes - for instance for *Airn*, it looks like there is 0.2 (20%?) expression of GFP in serum and somewhere around 2-3 in 2i. Clarification is required here or perhaps this can be illustrated in a different way.*

Response #10: We thank the referee for pointing this out. Both axes indeed show the % of GFP-expressing cells in the measured CRISPR cell populations. Reactivation in the serum conditions is indeed lower than 1%. We clarify that both axes show percentages in the legends text.

5. Line 239 should be 'extent' not 'extend'.

Response #11: we thank for pointing this out - we changed the text accordingly

6. In Figure 2 the authors note "CpG positions marked with x correspond to unaligned nucleotides". I am not sure what this means. Does the sequence quality fail at that point? Please explain this better.

Response #12: The "x" indicates nucleotides that failed the quality check during sequencing, i.e. Sanger sequencing returned Ns at this position. We have included this information in the figure legend.

Reviewer #2:

Remarks to the Author:

*In this work, Butz et al. investigate the maintenance of epigenetic states at imprinting control regions (ICRs). Integrating methylated and unmethylated sequences of three ICRs into a 'safe-harbor' locus in mouse ESCs, confirmed that ICRs can autonomously maintain their pre-established DNA methylation status, unlike other elements with similar GC content. The authors also observed that the state of DNA methylation is no longer preserved when shuffling the ICR sequence (even though CpGs remain intact), consistent with the notion that maintenance is dependent upon the binding of trans-acting factors (such as *Zfp57*). The authors then used the autonomous ICRs as reporters and performed a CRISPR/Cas9 screen to identify factors that are involved in DNA methylation maintenance. In addition to known factors like *Dnmt1*, *Uhrf1*, and *Zfp57*, they proposed additional hits and validated the involvement of two of them - *Atf7ip* and *Zmym2* in preserving DNA methylation and H3K9me3 at the three examined ICRs.*

Overall, the manuscript comprises two main parts: (i) studying the sequence composition (cis) of ICRs and (ii) identifying trans-acting factors that potentially regulate its inherent epigenetic bistability. While these are clearly fundamental questions, we find that, in its current state, the manuscript falls short in trying to address both topics.

We thank the referee for their comments, we have now included several new experiments to address both topics in more detail.

Main concerns:

1. The manuscript in its current form provides only a few new insights into cis determinants that asymmetrically regulate epigenetic states in ICRs. Presumably, a similar approach to Lienert et al. 2011 can be used here to uncover or refine binding motifs (such as for Zfp57), structural organization, and additional critical elements that can preserve the repressed state (even if demonstrated using a single ICR). Another outstanding issue is the minimal distance between the ICR and downstream promoter that is required to maintain repression. This can be addressed by integrating spacers with different GC content between the two elements. Finally, all tested ICRs are maternally imprinted and hence associated with specific gene promoters. This raises the question of whether paternally imprinted intergenic ICRs can similarly repress an adjacent promoter element?

Response #13:

We have now added additional results based on the experiments suggested by the reviewer. This includes a new series of experiments using smaller fragments of the Airn and of the H19 ICR. We observe that none of these smaller fragments can recapitulate the behavior we have observed for the full length ICRs, even if they contain Zfp57 sites or have similar CpG density to the original ICR. This suggests that there is no minimal region that is responsible for maintenance of epigenetic states, but rather multiple elements distributed across the entire ICR are required to work together. These new results are shown in Figure 2a and Sup. Fig 4b-d.

Based on this we focused our attention on Zfp57 binding sites in the Airn ICR and used a neuronal network-based interpretation of Zfp57 binding (BPNNet PMID: 33603233). By comparing the wild type sequence with sequences where we replaced 10 nucleotides in a sliding window approach, we observed that mutations in Zfp57 sites can influence nearby Zfp57 binding events (New Sup. Fig. 7 c-d). Finally, we showed that introducing the Zfp57 sites alone in a randomized Airn DNA background is sufficient to restore Zfp57 binding – this we show first using the in silico prediction (New Sup. Fig. 7e), then we confirm these results experimentally by introducing Airn-shuffle+Zfp57 into mouse stem cells (New Figure 2e)

Regarding investigating the distance between ICRs and promoters by using spacers. We agree that this is an important experiment to understand how the ICR is silencing the reporter in this context. But in our opinion, this cannot be addressed easily since this will not be only a question about the length of the linker, but also about the sequence content of the linker, which will differ depending on the linker size. Providing conclusive results from such experiments would require us to test a large space of different sequences of varying lengths and GC content and probably in all possible ICR/promoter combinations. This will not be possible in the allocated time and furthermore, also not the scope of the current manuscript which focuses on ICR sequences. Nevertheless, we think that this is a great suggestion, and we would investigate how we can address this question in the required depth in future projects.

We have now also included a reporter construct based on the paternally methylated H19 ICR in combination with the EF1a promoter. We observe that the H19 ICR can maintain repression of the EF1a reporter, suggesting that this is not limited to maternally methylated ICRs. These new results are shown in Sup. Fig. 8d and we thank the reviewer for this suggestion.

2. We appreciate that identifying 2 additional candidate genes that regulate the maintenance of ICR methylation is an important contribution to the field. However, current data does not go beyond screen validation. For example, KO of Atf7ip and Zmym2 in mESCs could imply that both of these genes directly bind Zfp57 or are involved in the recruitment of Kap1. This can be addressed using co-IP or other methods. Alternatively, any insights into the expression of Atf7ip and Zmym2 following implantation, in zygotes, and in germ cells, or the effects of KO on imprinting in vivo would significantly increase the impact of this report.

Response #14:

We thank the reviewer for the criticism and suggestions. We have now included several new experiments to address the role of ATF7IP and ZMYM2 in regulation of ICRs. Notably, have extended our genomics analysis to understand the genome-wide co-localization of ATF7IP, ZMYM2, ZFP57 and SETDB1 in mouse stem cells and furthermore their functional relevance for maintenance of epigenetic states at ICRs. We show now that all four factors co-localize to ICRs but show limited co-localization elsewhere – especially in the case of ZMYM2. Furthermore, we show that loss of both proteins leads to loss of DNA and H3K9me3 at the majority of tested ICRs. These data are presented in the new Figures 5 and 6.

In addition, we have included a biochemical characterization of protein-protein interactions. By using a set of experiments based on proximity biotinylation, the proteins that associate with ZFP57 could be detected. To our knowledge, this work identifies for the first-time factors that interact with ZFP57 on chromatin. This includes several known factors (KAP1, HP1, etc) but also previously unknown factors, including ATF7IP. The identification of ATF7IP suggests that this factor is part of the regulatory complex recruited to ICRs (New Figure 5d and Supplementary Figure 13d and Supplementary Table 4). These findings are in line with the ChIP-seq data that show a strong overlap between ZFP57/SETB1 sites and ATF7IP (new Figure 5b-c).

Finally, we interrogated the expression of *Atf7ip* and *Zmym2* during early development based on published datasets (PMID: 29281840) (Gao et al., 2017, <https://doi.org/10.1016/j.celrep.2017.11.111>). We see that the levels of ATF7IP and ZMYM2 are increased at the 2-cell stage and decline during the later stages of early development. We show this additional information in Supplemental Figure 13e. We agree that testing the direct involvement of these factors during mouse development would provide important insights into their role in setting up / maintaining imprints in vivo, however, this would be a major endeavor that would require large resources and time commitment, which would go beyond the scope of this manuscript. We hope the reviewer agrees with this point

Minor issues:

3. Figure 3. The authors should provide methylation data on the promoter regions integrated downstream of the ICRs to show that it indeed correlates with the methylation status of the ICR.

Response #15: This seems to be a misunderstanding in the design and in vitro-methylation step of our reporter constructs. When we generate the reporter cell lines, the entire construct including the ICR and the promoter are methylated in vitro and validated by methyl-sensitive restriction digest. This is later used for the RMCE-based integration of the entire reporter to the genome. Therefore, the integrated promoters immediately downstream of the ICRs have the same methylation status as the ICR. We have now rephrased the text to reduce any confusion.

4. One control that is missing is the integration of a methylated version of the promoter region - without the upstream ICR. This will provide information about the kinetics of demethylation when the repressive element is missing.

Response #16: We thank the reviewer for pointing this out. We have now repeated the experiments and included the methylated promoters with and without ICRs. These new results are shown in Figure 3b, indicating that the methylated promoters alone are not able to maintain the repressed state.

Furthermore, have performed a time course to address the dynamics of reactivation. In new Sup. Figure 8c we show the percentage of GFP positive cells in clonal populations obtained from individual RMCEs with various ICR-promoter combinations measured at 16, 23 and 30 days after integration. We show that the promoter-only constructs result in full expression of the GFP reporter (>90% of population GFP positive) already after 16 days, while reporters containing ICRs result in stable repression, indicating that in case of the PGK and EF1a promoters, the methylation status of the ICR dictates their activity.

*5. Can the authors include data validating KO of *Atf7ip* and *Zmym2* at the protein level? If this is not possible, validation at the genomic level should be provided*

Response #17: We have now included western blot analysis showing loss of ATF7IP or ZMYM2 in the respective KO clones. These new results can be found in Sup. Fig. 13f.

*6. Figure 3e. We find it surprising that KO of either *Uhrf1* or *Dnmt1* is not sufficient to de-repress the majority of reporter cells after 8 days in culture. Is that because the KO was done on the bulk population and not on validated KO clones? Can the authors explain this?*

Response #18: This is indeed because the GFP measurement was done on the bulk population after the CRISPR targeting and selection, resulting in a mix of cells harboring homozygous, heterozygous mutations, or homozygous wild type alleles. We show now the FACS profiles to indicate the bimodal distribution in the population (new Sup. Fig. 9a).

To overcome these limitations that stem from the varying success of generating KOs, we have used the novel DNA methylation inhibitor GSK-3484862 and show that already after 2 days treatment the reporters reactivate resulting in >90% of positive cells in the population. These new results are shown in Figure 3d and Supplemental Figure 9c.

Reviewer #3:

Remarks to the Author:

Imprint control regions (ICRs) are classic examples of epigenetic regulation, as genetically identical sequence exhibit differential DNA methylation states depending on the parent of origin. Moreover, these “bistable” loci (as the authors put it), can persist in somatic tissues throughout life, thus exhibiting remarkable epigenetic memory. Mouse ESCs are a pertinent model to study ICR regulation: recent studies have shown that epigenetic memory is less stable in ESCs than in differentiated cell types (e.g., <https://www.biorxiv.org/content/10.1101/2021.05.11.443595v1>); however ICRs are an exception, as the factors that protect the regions from losing DNA methylation during epigenetic reprogramming are likely present. In this study, Butz et al. take advantage of this fact to determine the properties of DNA Methylation-based ICR control, and perform a CRISPR-based screen to discover important factors in this process.

In fact, the manuscript is essentially two different studies. In the first part, the authors utilized the recombinase-mediated cassette exchange (RMCE) system to efficiently integrate a set of ICR sequences into the “neutral” β globin locus. This is a strategy that has been used with high success in previous studies from the Schuebeler lab, where the senior author from this study did his postdoctoral work. I found the experiments to be well-executed and convincing: pre-methylated ICRs tend to stay methylated, while unmethylated versions of the same sequences remain DNA methylation free—although it should be noted that in some cases de novo DNA methylation occurs at the unmethylated sequences, and this is stably maintained (Supp. Fig. 1d-f); this is a minor point, but is not commented on in the text.

The DNA methylation maintenance does not occur on sequences with similar length, GC content, and CpG density. Moreover, scrambling the sequence between CpGs abolishes DNA methylation maintenance at ICRs. The authors state that this shows that the DNA sequence—in addition to the DNA methylation—that confers its epigenetic properties. These experiments are a nice way to demonstrate this fact, but it is not a novel finding. It has been known for some time that ZFP57 (and more recently ZFP445) recognize both genetic and epigenetic information. Therefore, this is more of a confirmation than a discovery. And as mentioned, given ESCs exhibit epigenetic plasticity, it is again rather expected CpG-rich regions absent these key genetic motifs would not maintain DNA methylation. My presumption is if this same experiment was performed in a differentiated cell-type, the methylated sequences very likely might maintain the mark.

The second part of the paper was a CRISPR based genetic screen taking advantage of the ICR epigenetic stability. By placing pre-methylated ICRs upstream of a strong promoter and a reporter, in some cases the reporter would be silenced. Therefore, mutagenesis screens with various ICR/reporter combinations could identify key regulators of ICR maintenance. Indeed, DNA methylation maintenance factors Dnmt1 and Uhrf1 were enriched in the screens, as well as Zfp57. The authors focused on two more factors: ATF7IP and ZMYM2. Beyond a ChIP-seq experiment, there was very little characterization of the factors, however. The superficiality of this portion of the paper made it feel somewhat shoe-horned in.

In summary, while the first part of the paper is strong, it suffers from lack of novelty. The second part could potentially lead to interesting directions with more in depth experiments, but those were not performed. Below I have added some minor comments that could strengthen the paper for a near-term submission:

Comments

- If the ESCs are differentiated, and the RMCE experiment is re-performed, is the genetic component of the ICRs less important? For example, would the shuffled Airn ICR maintain its methylation better out of an ESC context. I think this would be important to show how important genetic differences would be for ESCs/epigenetic reprogramming, but the epigenetic signature is more important at later stages of differentiation/development.

Response #19: Due to the limitations of our RMCE system, we cannot introduce the ICRs to the RMCE acceptor site in differentiated cells. This requires a selection step to enrich for cells that successfully have integrated the DNA. In ES cells, this is done by limited dilution and clonal selection for ca 14 days, followed by expansion of clones to obtain enough material for experiments (another 7 days). This is not possible to perform in our ES to post-mitotic neuron differentiation system due to lack of cell growth.

However, we agree that this is a very important question. Especially since ZFP57 is expressed predominantly in the germline and ES cells. Therefore, we now address this question by making use of differentiated murine erythroleukemia cell lines (MEL) cell that contain a similar RMCE system. Here we integrated the shuffled Airn ICR as methylated and unmethylated DNA. In contrast to ES cells, DNA methylation states are faithfully maintained at the shuffled Airn in MEL cells, suggesting that indeed the epigenetic signature is sufficient for its maintenance and that sequence-specific factors are not required at later stages of development. We thank the reviewer for this suggestion, and we have included these results in Figure 2f.

• 1b - Tcl is not apparently protected from de novo methylation (45.9% methylated in Sssl -), and then the methylated sequence gets eroded. What is the nature of this erosion? Do molecules look methylated and unmethylated, or it exhibits a "salt and pepper". This jumped out to me because it seems to fit an intermediate category of DNA methylation not discussed.

Response #20: Why the Tcl sequence is gaining DNA methylation is not fully clear and has not been part of this study. Please note that the results obtained using the unmethylated promoters shown in Figure 1 were from Lienert et al 2011 (as indicated in the legend). We can only speculate that the CpG density of this fragment was not sufficient to ensure complete protection from methylation or that TF binding sites in the vicinity were missing.

• Supp. Fig. 1d-f: The de novo + maintenance property of some of the unmethylated ICRs should be discussed. Is this just a random event that occurs upon integration at the RMCE site?

Response #21: As requested, we now mention the de novo methylation observed at some of the unmethylated ICRs in the text. Similar to the Tcl promoter, we can only speculate that unmethylated ICRs are more context-dependent than methylated ICRs. For example, several ICRs are located downstream of promoters, and since these were not included in the construct, lack of transcription from these promoters could reduce the protection from de novo methylation. Furthermore, active ICRs are in contact with nearby regulatory regions via looping and since we integrate the ICRs at a different location, these interactions are missing, leading to reduced protection from de novo methylation. However, we do not have any experimental evidence to back up these speculations as our main focus in this manuscript was on the methylated ICRs.

• What promoters were finally used for the three ICRs in the screen? This is minor, but it is not clear in the manuscript

Response #22: We thank the reviewer for pointing this out. We indicate now in the text and figures which promoters were used in every screen.

• The characterization of ZMYM2 (in particular) and ATF7IP could have gone much deeper. Here are some quick ideas:

- What is the global methylation state in the mutants? This could be done by a LUMA assay, or potentially a whole-genome sequencing approach (e.g., RRBS or WGBS)
- What is the impact of methylation maintenance of ICRs in a Zmym2 mutant? If these cells grow slowly, this could be done with a degen system for example.

Response #23: We thank the reviewer for these suggestions. We now provide genome-wide bisphite sequencing data to address the impact of ATF7IP and ZMYM2 loss in mouse stem cells. We see that global methylation is not strongly reduced as the majority of genomic elements are above 70 % mCG. However, we observe that the methylation at the majority of ICRs is strongly reduced/depleted in ATF7IP and ZMYM2 KO cells. We show these new results in Figures 6a and Supplementary Figure 14.

- The ChIP seq data should be analyzed more thoroughly. For starters, the binding status at all ICRs should be discussed.

Response #24: We now provide in depth analysis of ATF7IP and ZMYM2 binding to all ICR and genome-wide. We observe that both proteins, despite having distinct binding sites outside of ICRs, both bind to all tested ICR regions, with the only exceptions at H13 and H19. In addition, we analyzed the genome-wide binding of both factors and compared it to binding of SETDB1 and ZFP57 outside of ICRs. We also provide H3K9me3 and DNA methylation analysis in KO ES cells for the respective proteins. These new results are shown in new Figures 5-6

and Sup. Figures 13 - 15. We hope the reviewer agrees that these additional datasets and results strengthen our initial observations.

Decision Letter, first revision:

Our ref: NG-A57780R

21st Jun 2022

Dear Tuncay,

Thank you for submitting your revised manuscript entitled "DNA sequence and chromatin modifiers cooperate to confer epigenetic bistability at imprinting control regions" (NG-A57780R). It has now been seen by two of the original referees and their comments are below. Unfortunately, reviewer #1 was unable to submit a timely report. We have now decided to proceed with a decision.

The reviewers find that the paper has improved in revision, and therefore we'll be happy in principle to publish it in Nature Genetics, pending minor revisions to satisfy the referees' final requests and to comply with our editorial and formatting guidelines.

The current version of your manuscript is in a PDF format, please email us (genetics@us.nature.com) a copy of the file in an editable format (Microsoft Word) -- we can not proceed with PDFs at this stage.

We will then be performing detailed checks on your paper and will send you a checklist detailing our editorial and formatting requirements soon. Please do not upload the final materials and make any revisions until you receive this additional information from us.

Thank you again for your interest in Nature Genetics. Please do not hesitate to contact me if you have any questions.

Congratulations!

Sincerely,

Tiago

Tiago Faial, PhD
Senior Editor
Nature Genetics
<https://orcid.org/0000-0003-0864-1200>

Reviewer #2 (Remarks to the Author):

I commend the authors for addressing the main and critical comments to a considerable extent. Overall, this study includes extensive work shedding additional light on the interplay between DNA sequence, epigenetic modifications, and trans-acting factors to maintain parent-specific epialleles. Readers of this manuscript would benefit if the authors discuss the following points:

(1) The cofactor ATF7IP was previously associated with Setdb1, which is known to be recruited by TRIM28, that in turn binds to chromatin via KRAB Zinc finger proteins. The findings here, along with the additional data for the genome wide occupancy of ATF7IP and SETDB1, nicely confirm these associations as part of the established complex for ZFP57-TRIM28-SETDB1-ATF7IP.

Mechanistically, however, it is still unclear how the knockout of any of the factors in the complex leads to demethylation in the face of intact housekeeping activity of Uhrf1 and Dnmt1. Thus, the unique epigenetic landscape associated with mESCs in culture may not faithfully represent the maintenance activity in vivo.

Along these lines, germline and zygote knockout of Zfp57, together with experiments showing its preferential binding to a methylated motif, were key for positioning this factor as a central player in the maintenance of ICRs. As stated by the authors, these experiments are beyond the scope of the current study. Yet without this confirmation, conclusions that can be made regarding the role of these novel factors in maintaining imprints – are lacking. This caveat should be clearly stated.

(2) A similar approach for using imprinted DMRs as DNA methylation reporters and its implementation for screening trans-acting factors that regulate methylation were previously published (PMID's: 33833093, 26406378) yet are not mentioned by the authors. It would be interesting to discuss the different principles used in both systems (somatic vs. ICRs) and their potential impact on identifying cis and trans effects.

Reviewer #3 (Remarks to the Author):

The authors have addressed all of my concerns, and have also added a significant amount of experimental data to the study. To reaffirm: I think this is a very solid study that has a lot of pertinent and useful data for the DNA methylation / imprinting field(s). I would not say that the study provides major advancements into our understanding of imprint maintenance mechanism, but the findings presented here will be of general interest.

Author Rebuttal, first revision:

Point by point response to Reviewee's comments:

10Reviewer #1:

None

Reviewer #2:

Remarks to the Author:

I commend the authors for addressing the main and critical comments to a considerable extent. Overall, this study includes extensive work shedding additional light on the interplay between DNA sequence, epigenetic modifications, and trans-acting factors to maintain parent-specific epialleles. Readers of this manuscript would benefit if the authors discuss the following points:

(1) The cofactor ATF7IP was previously associated with Setdb1, which is known to be recruited by TRIM28, that in turn binds to chromatin via KRAB Zinc finger proteins. The findings here, along with the additional data for the genome wide occupancy of ATF7IP and SETDB1, nicely confirm these associations as part of the established complex for ZFP57-TRIM28-SETDB1-ATF7IP.

Mechanistically, however, it is still unclear how the knockout of any of the factors in the complex leads to demethylation in the face of intact housekeeping activity of Uhrf1 and Dnmt1. Thus, the unique epigenetic landscape associated with mESCs in culture may not faithfully represent the maintenance activity in vivo.

Along these lines, germline and zygote knockout of Zfp57, together with experiments showing its preferential binding to a methylated motif, were key for positioning this factor as a central player in the maintenance of ICRs. As stated by the authors, these experiments are beyond the scope of the current study. Yet without this confirmation, conclusions that can be made regarding the role of these novel factors in maintaining imprints – are lacking. This caveat should be clearly stated.

Response 1 – We agree with this comment and have made the following statement in the manuscript: “If and how these two factors contribute to maintenance of imprints during zygote formation and early development remains to be tested. In ESCs however, their absence resulted in impaired maintenance fidelity and sporadic loss of H3K9me3 at multiple ICRs, independent of their parental origin.”

(2) A similar approach for using imprinted DMRs as DNA methylation reporters and its implementation for screening trans-acting factors that regulate methylation were previously published (PMID's: 33833093, 26406378) yet are not mentioned by the authors. It would be interesting to discuss the different principles used in both systems (somatic vs. ICRs) and their potential impact on identifying cis and trans effects.

Response 2 – We have now included these references and discuss the difference between the used systems in the Discussion: “In contrast to previous strategies that employed the *Snrpn* promoter to report changes in methylation at endogenous gene promoters^{49,50}, our cell lines directly report regulatory changes at the introduced ICRs.”

Reviewer #3:

Remarks to the Author:

The authors have addressed all of my concerns, and have also added a significant amount of experimental data to the study. To reaffirm: I think this is a very solid study that has a lot of pertinent and useful data for the DNA

methylation / imprinting field(s). I would not say that the study provides major advancements into our understanding of imprint maintenance mechanism, but the findings presented here will be of general interest.

Response 3 – We thank the reviewer for their comment.

Final Decision Letter:

In reply please quote: NG-A57780R1 Baubec

19th Sep 2022

Dear Tuncay,

I am delighted to say that your manuscript "DNA sequence and chromatin modifiers cooperate to confer epigenetic bistability at imprinting control regions" has been accepted for publication in an upcoming issue of Nature Genetics.

Your paper will be published online after we receive your corrections and will appear in print in the next available issue. You can find out your date of online publication by contacting the Nature Press Office (press@nature.com) after sending your e-proof corrections. Now is the time to inform your Public Relations or Press Office about your paper, as they might be interested in promoting its publication. This will allow them time to prepare an accurate and satisfactory press release. Include your manuscript tracking number (NG-A57780R1) and the name of the journal, which they will need when they contact our Press Office.

12Before your paper is published online, we shall be distributing a press release to news organizations worldwide, which may very well include details of your work. We are happy for your institution or funding agency to prepare its own press release, but it must mention the embargo date and Nature Genetics. Our Press Office may contact you closer to the time of publication, but if you or your Press Office have any enquiries in the meantime, please contact press@nature.com.

Please note that *Nature Genetics* is a Transformative Journal (TJ). Authors may publish their research with us through the traditional subscription access route or make their paper immediately open access through payment of an article-processing charge (APC). Authors will not be required to make a final decision about access to their article until it has been accepted. [Find out more about Transformative Journals](https://www.springernature.com/gp/open-research/transformative-journals)

Authors may need to take specific actions to achieve [compliance](https://www.springernature.com/gp/open-research/funding/policy-compliance-faqs) with funder and institutional open access mandates. If your research is supported by a funder that requires immediate open access (e.g. according to [Plan S principles](https://www.springernature.com/gp/open-research/plan-s-compliance)) then you should select the gold OA route, and we will direct you to the compliant route where possible. For authors selecting the subscription publication route, the journal's standard licensing terms will need to be accepted, including [self-archiving and license to publish](https://www.nature.com/nature-portfolio/editorial-policies/self-archiving-and-license-to-publish). Those licensing terms will supersede any other terms that the author or any third party may assert apply to any version of the manuscript.

Please note that Nature Portfolio offers an immediate open access option only for papers that were first submitted after 1 January, 2021.

Sincerely,

Tiago

Tiago Faial, PhD
Chief Editor
Nature Genetics
<https://orcid.org/0000-0003-0864-1200>